# DreamWaltz: Make a Scene with Complex 3D Animatable Avatars

**Yukun Huang**[1,2*†]**, Jianan Wang**[1*‡]**, Ailing Zeng**[1]**, He Cao**[1]**, Xianbiao Qi**[1]**, Yukai Shi**[1]**,
Zheng-Jun Zha**[2]**, Lei Zhang**[1]

[1]International Digital Economy Academy (IDEA)
[2]University of Science and Technology of China

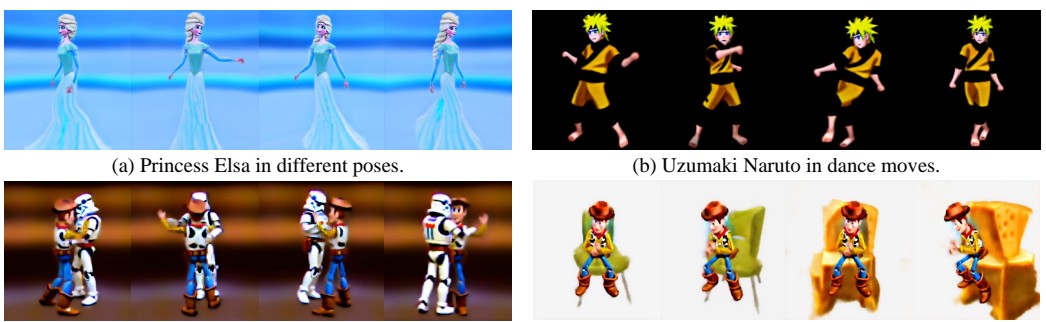

(a) Princess Elsa in different poses.      (b) Uzumaki Naruto in dance moves.

(c) Woody dances with Stormtrooper.      (d) Woody sits on: (**Left**) a chair; (**Right**) a chair made of cheese.

Figure 1: DreamWaltz is a text-to-3D-avatar generation framework, which can (a, b) create complex 3D animatable avatars from texts, (c, d) ready for 3D scene composition with diverse interactions.

## Abstract

We present DreamWaltz, a novel framework for generating and animating complex 3D avatars given text guidance and parametric human body prior. While recent methods have shown encouraging results for text-to-3D generation of common objects, creating high-quality and animatable 3D avatars remains challenging. To create high-quality 3D avatars, DreamWaltz proposes 3D-consistent occlusion-aware Score Distillation Sampling (SDS) to optimize implicit neural representations with canonical poses. It provides view-aligned supervision via 3D-aware skeleton conditioning which enables complex avatar generation without artifacts and multiple faces. For animation, our method learns an animatable 3D avatar representation from abundant image priors of diffusion model conditioned on various poses, which could animate complex non-rigged avatars given arbitrary poses without retraining. Extensive evaluations demonstrate that DreamWaltz is an effective and robust approach for creating 3D avatars that can take on complex shapes and appearances as well as novel poses for animation. The proposed framework further enables the creation of complex scenes with diverse compositions, including avatar-avatar, avatar-object and avatar-scene interactions. See https://dreamwaltz3d.github.io/ for more vivid 3D avatar and animation results.

## 1 Introduction

The creation and animation of 3D digital avatars are essential for various applications, including film and cartoon production, video game design, and immersive media such as AR and VR. However,

---

[*]Equal contribution.
[†]Work done during an internship at IDEA.
[‡]Corresponding author.

37th Conference on Neural Information Processing Systems (NeurIPS 2023).

traditional techniques for constructing such intricate 3D models are costly and time-consuming, requiring thousands of hours from skilled artists with extensive aesthetics and 3D modeling knowledge. In this work, we seek a solution for 3D avatar generation that satisfies the following desiderata: (1) easily controllable over avatar properties through textual descriptions; (2) capable of producing high-quality and diverse 3D avatars with complex shapes and appearances; (3) the generated avatars should be ready for animation and scene composition with diverse interactions.

The advancement of deep learning methods has enabled promising methods which can reconstruct 3D human models from monocular images [36, 45] or videos [44, 14, 46, 41, 12, 30]. Nonetheless, these methods rely heavily on the strong visual priors from image/video and human body geometry, making them unsuitable for generating creative avatars that can take on complex and imaginative shapes or appearances. Recently, integrating 2D generative models into 3D modeling [31, 18, 10] has gained significant attention to make 3D digitization more accessible, reducing the dependency on extensive 3D datasets. However, creating animatable 3D avatars remains challenging: Firstly, avatars often require intricate and complex details for their appearance (e.g., loose cloth, diverse hair, and different accessories); secondly, avatars have articulated structures where each body part is able to assume various poses in a coordinated and constrained way; and thirdly, avatar changes shape and texture details such as creases when assuming different poses, making animation for complex avatars extremely challenging. As a result, while DreamFusion [31] and subsequent methods [18, 5] have demonstrated impressive results on text-guided creation of stationary everyday objects, they lack the proper constraints to enforce consistent 3D avatar structures and appearances, which presents significant challenges to producing intricate shapes, appearances and poses for 3D avatars, let alone for animation.

In this paper, we present DreamWaltz, a framework for generating high-quality 3D digital avatars from text prompts utilizing human body prior of shapes and poses, ready for animation and composition with diverse avatar-avatar, avatar-object and avatar-scene interactions. DreamWaltz employs a trainable Neural Radiance Field (NeRF) as the 3D avatar representation, a pre-trained text-and-skeleton-conditional diffusion model [48] for shape and appearance supervision, and SMPL models [2] for extracting 3D-aware posed-skeletons. Our method enables high-quality avatar generation with 3D-consistent SDS, which resolves the view disparity between the diffusion model's supervision and NeRF's rendering. By training an animatable NeRF with diffusion supervision conditioned on human pose prior, we can deform the generated non-rigged avatar to arbitrary poses for realistic test-time animation without retraining.

The key contributions of DreamWaltz lie in four main aspects:

- We propose a novel text-to-avatar generation framework named DreamWaltz, which is capable of creating animatable 3D avatars with complex shapes and appearances.

- For avatar creation, we propose a SMPL-guided 3D-consistent Score Distillation Sampling strategy with occlusion culling, enabling the generation of high-quality avatars, e.g., avoiding the Janus (multi-face) problem and limb ambiguity.

- We propose to learn an animatable NeRF representation from diffusion model and human pose prior, which enables the animation of complex avatars. Once trained, we can animate the created avatar with any pose sequence without retraining.

- Experiments show that DreamWaltz is effective in creating high-quality and animatable avatars, ready for scene composition with diverse interactions across avatars and objects.

## 2   Related Work

**Text-guided image generation.**   Recently, there have been significant advancements in text-to-image models such as GLIDE [26], unCLIP [33], Imagen [35], and Stable Diffusion [34], which enable the generation of highly realistic and imaginative images based on text prompts. These generative capabilities have been made possible by advancements in modeling, such as diffusion models [6, 39, 27], and the availability of large-scale web data containing billions of image-text pairs [37, 38, 4]. These datasets encompass a wide range of general objects, with significant variations in color, texture, and camera viewpoints, providing pre-trained models with a comprehensive understanding of general objects and enabling the synthesis of high-quality and diverse objects.

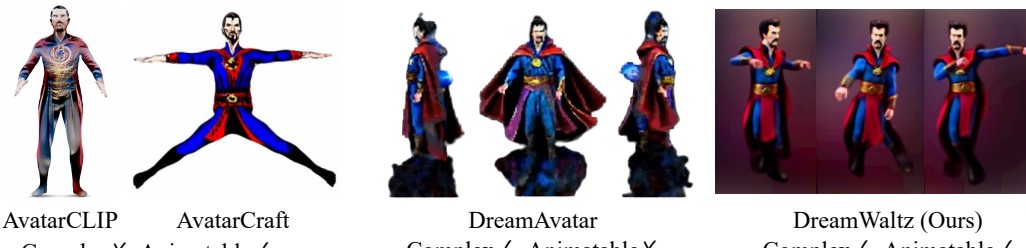

| AvatarCLIP | AvatarCraft | | DreamAvatar | | DreamWaltz (Ours) |
| Complex✗ | Animatable✓ | | Complex✓ | Animatable✗ | Complex✓ | Animatable✓ |

Figure 2: Comparison of text-driven 3D avatar generation methods, including: AvatarCLIP [8], AvatarCraft [13], DreamAvatar [3] and DreamWaltz (Ours). AvatarCLIP and AvatarCraft assume strong SMPL constraints, which makes it straightforward for the generated avatars to align with SMPL for animation. But due to the constraints, the avatars *cannot take on complex shapes and appearances*; With weak SMPL constraints, DreamAvatar struggles with wrong avatar geometry and *requires retraining* for each pose adjustment. Different from existing methods, DreamWaltz enables complex and animatable 3D avatar generation benefiting from the proposed SMPL-guided 3D-consistent SDS and deformation learning from human pose prior.

Furthermore, recent works [48, 9, 15] have explored incorporating additional conditioning, such as depth maps and human skeleton poses, to generate images with more precise control.

**Text-guided 3D generation.** Dream Fields [11] and CLIPmesh [23] were groundbreaking in their utilization of CLIP [32] to optimize an underlying 3D representation, aligning its 2D renderings with user-specified text prompts, without necessitating costly 3D training data. However, this approach tends to result in less realistic 3D models since CLIP only provides discriminative supervision for high-level semantics. In contrast, recent works have demonstrated remarkable text-to-3D generation results by employing powerful text-to-image diffusion models as a robust 2D prior for optimizing a trainable NeRF with Score Distillation Sampling (SDS) [31, 18, 5, 10]. Nonetheless, the produced geometry and texture are static, blurry and usually lack intricate details necessary for avatar creation.

**Text-guided 3D avatar generation.** Avatar-CLIP [8] starts with initializing 3D human geometry via a shape VAE network and subsequently employs CLIP [32] for shape sculpting and texture generation. To animate the generated 3D avatar, they propose a CLIP-guided reference-based motion synthesis method. However, this approach tends to produce less realistic and oversimplified 3D models due to the limited guidance provided by CLIP, which primarily focuses on high-level semantic discrimination. Concurrent to our work, DreamAvatar [3] and AvatarCraft [13] both utilize pre-trained text-to-image diffusion models and SMPL models as shape prior for avatar generation. While DreamAvatar focuses on producing static posed-3D avatars which are incapable of animation, AvatarCraft generates higher-quality avatars via coarse-to-fine and multi-box training and enables animation via local transformation between the template mesh and the target mesh based on SMPL models. We summarize the key differences between our work and related works in Fig. 2.

## 3 Method

### 3.1 Preliminary

**Text-to-3D generation.** Recent methods [31, 18, 21] have shown encouraging results on text-to-3D generation of common objects by integrating three essential components:

*(1) Neural Radiance Fields* (NeRF) [25, 1, 22] is commonly adopted as the 3D representation for text-to-3D generation [40, 18], parameterized by a trainable MLP. For rendering, a batch of rays $\mathbf{r}(k) = \mathbf{o} + k\mathbf{d}$ are sampled based on the camera position $\mathbf{o}$ and direction $\mathbf{d}$ on a per-pixel basis. The MLP takes $\mathbf{r}(k)$ as input and predicts density $\tau$ and color $c$. The volume rendering integral is then approximated using numerical quadrature to yield the final color of the rendered pixel:

$$\hat{C}_c(\mathbf{r}) = \sum_{i=1}^{N_c} \Omega_i \cdot (1 - \exp(-\tau_i \delta_i)) c_i,$$

where $N_c$ is the number of sampled points on a ray, $\Omega_i = \exp(-\sum_{j=1}^{i-1} \tau_j \delta_j)$ is the accumulated transmittance, and $\delta_i$ is the distance between adjacent sample points.

*(2) Diffusion models* [7, 27] which have been pre-trained on extensive image-text datasets [33, 35, 40] provide a robust image prior for supervising text-to-3D generation. Diffusion models learn to estimate the denoising score $\nabla_{\mathbf{x}} \log p_{\text{data}}(\mathbf{x})$ by adding noise to clean data $\mathbf{x} \sim p(\mathbf{x})$ (forward process) and learning to reverse the added noise (backward process). Noising the data distribution to isotropic Gaussian is performed in $T$ timesteps, with a pre-defined noising schedule $\alpha_t \in (0, 1)$ and $\bar{\alpha}_t := \prod_{s=1}^{t} \alpha_s$, according to:

$$\mathbf{z}_t = \sqrt{\bar{\alpha}_t} \mathbf{x} + \sqrt{1 - \bar{\alpha}_t} \boldsymbol{\epsilon}, \text{ where } \boldsymbol{\epsilon} \sim \mathcal{N}(\mathbf{0}, \mathbf{I}).$$

In the training process, the diffusion models learn to estimate the noise by

$$\mathcal{L}_t = \mathbb{E}_{\mathbf{x}, \boldsymbol{\epsilon} \sim \mathcal{N}(\mathbf{0}, \mathbf{I})} \left[ \left\| \boldsymbol{\epsilon}_\phi \left( \mathbf{z}_t, t \right) - \boldsymbol{\epsilon} \right\|_2^2 \right].$$

Once trained, one can estimate $\mathbf{x}$ from noisy input and the corresponding noise prediction.

*(3) Score Distillation Sampling* (SDS) [31, 18, 21] is a technique introduced by DreamFusion [31] and extensively employed to distill knowledge from a pre-trained diffusion model $\boldsymbol{\epsilon}_\phi$ into a differentiable 3D representation. For a NeRF model parameterized by $\boldsymbol{\theta}$, its rendering $\mathbf{x}$ can be obtained by $\mathbf{x} = g(\boldsymbol{\theta})$ where $g$ is a differentiable renderer. SDS calculates the gradients of NeRF parameters $\boldsymbol{\theta}$ by,

$$\nabla_{\boldsymbol{\theta}} \mathcal{L}_{\text{SDS}}(\phi, \mathbf{x}) = \mathbb{E}_{t, \boldsymbol{\epsilon}} \left[ w(t) (\boldsymbol{\epsilon}_\phi(\mathbf{x}_t; y, t) - \boldsymbol{\epsilon}) \frac{\partial \mathbf{z}_t}{\partial \mathbf{x}} \frac{\partial \mathbf{x}}{\partial \boldsymbol{\theta}} \right], \tag{1}$$

where $w(t)$ is a weighting function that depends on the timestep $t$ and $y$ denotes the given text prompt.

**SMPL** [19] is a 3D parametric human body model with a vertex-based linear deformation model, which decomposes body deformation into identity-related and pose-related shape deformation. It contains $N = 6,890$ vertices and $K = 24$ keypoints. Benefiting from its efficient and expressive human motion representation ability, SMPL has been widely used in human motion-driven tasks [8, 47, 20]. SMPL parameters include a 3D body joint rotation $\xi \in \mathbb{R}^{K \times 3}$, a body shape $\beta \in \mathbb{R}^{10}$, and a 3D global scale and translation $t \in \mathbb{R}^3$.

Formally, constructing a rest pose $T(\beta, \xi)$ involves combining the mean template shape $\bar{T}$ from the canonical space, the shape-dependent deformations $B_S(\beta) \in \mathbb{R}^{3N}$, and the pose-dependent deformations $B_P(\xi) \in \mathbb{R}^{3N}$ to relieve artifacts in a standard linear blend skinning (LBS) [24] by,

$$T(\beta, \xi) = \bar{T} + B_S(\beta) + B_P(\xi).$$

To map the SMPL parameters $\beta, \xi$ to a triangulated mesh, a function $M$ is adopted to combine the rest pose mesh $T(\beta, \xi)$, the corresponding keypoint positions $\mathcal{J}(\beta) \in \mathbb{R}^{3K}$, pose $\xi$, and a set of blend weights $\mathcal{W} \in \mathbb{R}^{N \times K}$ via a LBS function as,

$$M(\beta, \xi) = \text{W}(T(\beta, \xi), \mathcal{J}(\beta), \xi, \mathcal{W}).$$

To obtain the corresponding vertex under the observation pose $\mathbf{v}_o$, an affine deformation $\mathcal{G}_k(\xi, j_k)$ with skinning weight $w_k$ is used to transform the $k_{th}$ keypoint $j_k$ from the canonical pose to the observation pose as,

$$\mathbf{v}_o = \sum_{k=1}^{K} w_k \mathcal{G}_k(\xi, j_k). \tag{2}$$

## 3.2 DreamWaltz: A Text-to-Avatar Generation Framework

### 3.2.1 Creating a Canonical Avatar

DreamWaltz employs a trainable NeRF as the 3D avatar representation. It leverages SMPL prior in two ways: (1) initializing NeRF; (2) extracting 3D-aware and occlusion-aware posed-skeletons to condition ControlNet [48] for 3D-consistent Score Distillation Sampling. Fig 3 (a) illustrates our method on how to create a canonical avatar.

**Stage I: Canonical Avatar Creation**

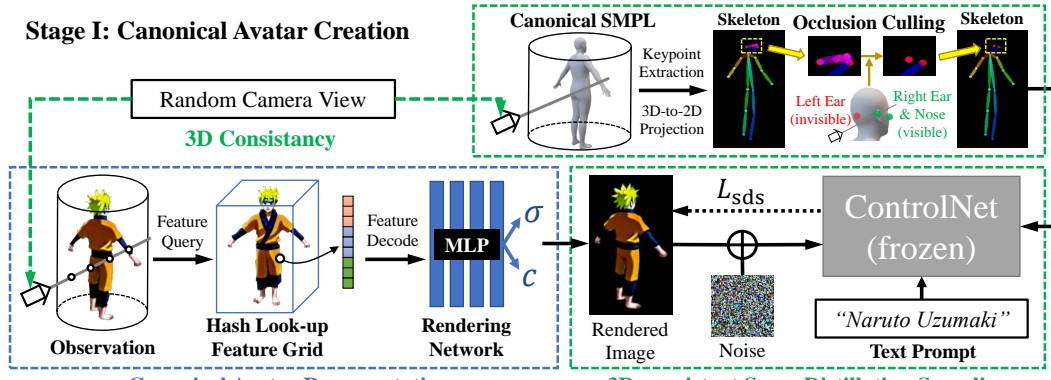

(a) **Canonical Avatar Creation.** We randomly sample a camera viewpoint to render both the 3D avatar and an SMPL model. The extracted SMPL 3D keypoints are projected to a 2D skeleton image with occlusion culling. The skeleton image is by construction 3D-consistent with the rendered avatar. We then utilize ControlNet [48] conditioned on the text prompt and the skeleton image instead of the original SD [34].

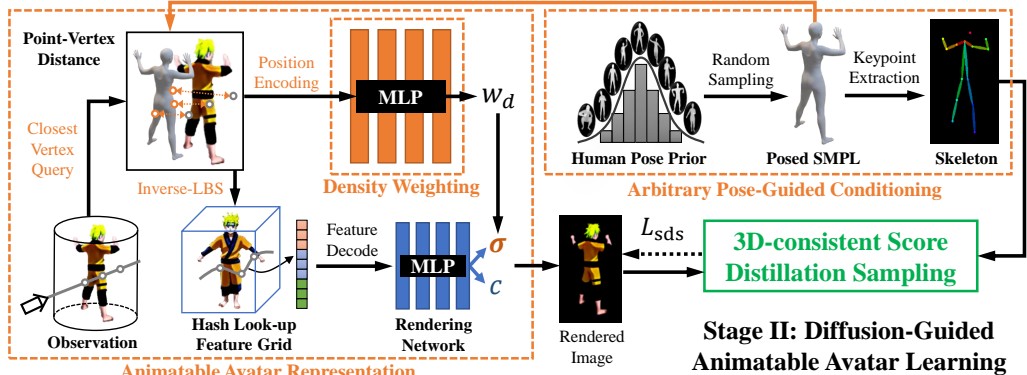

(b) **Animatable Avatar Learning.** We randomly sample viable poses from off-the-shelf VPoser [29] for SMPL model to condition ControlNet [48] and to learn a generalizable density weighting function to refine vertex-based pose transformation, enabling animation of complex avatars.

Figure 3: Illustration of our framework for canonical and animatable avatar creation. (a) shows how to create a canonical avatar from text with 3D-consistent occlusion-aware Score Distillation Sampling, and (b) demonstrates how to further learn an animatable avatar with sampled human pose prior.

**SMPL-guided initialization.** To speed up the NeRF optimization and to provide a reasonable initial input for retrieving informative supervision from the diffusion model, we pre-train NeRF based on an SMPL mesh. The SMPL model could be in the canonical pose as adopted in our method to avoid self-occlusion or in any chosen pose for posed-avatar creation [3]. Specifically, we render the silhouette image $x_{sil}$ of the SMPL model given a randomly sampled viewpoint and minimize the MSE loss between the NeRF-rendered image $x$ and the silhouette image $x_{sil}$. Note that our NeRF renders images in the latent space of Stable Diffusion [34], so it is necessary to use the VAE-based image encoder to transform silhouette images into the latent space for the loss calculation. We empirically found that SMPL-guided NeRF initialization significantly improves the geometry and the convergence speed for avatar generation.

**3D-consistent Score Distillation Sampling.** Vanilla SDS [40, 43, 18] as previously introduced in Sec. 3.1 utilizes view-dependent prompt augmentations such as "front view of ..." for diffusion model to provide crucial 3D view-consistent supervision. However, this prompting strategy cannot guarantee precise view consistency, leaving the disparity between the viewpoint of the diffusion model's supervision image and NeRF's rendering image unresolved. Such inconsistency causes quality issues for 3D generation, such as blurriness and the Janus (multi-face) problem. Inspired by recent works of controllable image generation [48, 15], we propose to utilize additional 3D-aware conditioning images to improve SDS for 3D-consistent NeRF optimization. Specifically, additional

conditioning image $c$ is injected to Eq. 1 for SDS gradient computation:

$$\nabla_{\boldsymbol{\theta}}\mathcal{L}_{\text{SDS}}(\phi, \mathbf{x}) = \mathbb{E}_{t,\boldsymbol{\epsilon}}\left[w(t)(\boldsymbol{\epsilon}_\phi(\mathbf{x}_t; y, t, \boxed{c}) - \boldsymbol{\epsilon})\frac{\partial \mathbf{z}_t}{\partial \mathbf{x}}\frac{\partial \mathbf{x}}{\partial \boldsymbol{\theta}}\right], \tag{3}$$

where conditioning image $c$ can be one or a combination of skeletons, depth maps, normal maps and etc. In practice, we choose skeletons as the conditional image type because they provide minimal image structure priors and enable complex avatar generation. In order to acquire 3D-consistent supervision, the conditioning image's viewpoint should be in sync with NeRF's rendering viewpoint. To achieve this for avatar generation, we use human SMPL models to produce conditioning images.

**Occlusion culling.** The introduction of 3D-aware conditional images can enhance the 3D consistency in the SDS optimization process. However, the effectiveness is constrained by the adopted diffusion model [48] on its interpretation of the conditional images. As shown in Fig. 8 (a), we provide a back-view skeleton map as the conditional image to ControlNet [48] and perform text-to-image generation, but a frontal face still appears in the generated image. Such defects bring problems such as multiple faces and unclear facial features to 3D avatar generation. To this end, we propose to use occlusion culling algorithms [28] in computational graphics to detect whether facial keypoints are visible from the given viewpoint and subsequently remove them from the skeleton map if considered invisible. Body keypoints remain unaltered because they reside in the SMPL mesh, and it is difficult to determine whether they are occluded without introducing new priors.

### 3.2.2 Learning an Animatable Avatar

Fig. 3 (b) illustrates our framework for generating animatable 3D avatars. In the training process, we randomly sample SMPL models of viable poses from VPoser [29] to condition ControlNet [48] and to learn a generalizable density weighting function to refine vertex-based pose transformation, enabling animation of complex avatars. At test time, DreamWaltz is capable of creating an animation based on arbitrary motion sequences *without requiring further pose-by-pose retraining*.

**SMPL-guided avatar articulation.** Referring to Sec. 3.1, SMPL defines a vertex transformation from observation space to canonical space according to Eq. 2. In this work, we use SMPL-guided transformation to achieve NeRF-represented avatar articulation. More concretely, for each sampled point $\mathbf{p}$ on a NeRF ray, we find its closest vertex $\mathbf{v}_c$ based on a posed SMPL mesh. We then utilize the transformation matrix $\mathbf{T}_{\text{skel}}$ of $\mathbf{v}_c$ to project $\mathbf{p}$ to the canonical space feature grid for feature querying and subsequent volume rendering. When $\mathbf{p}$ is close to $\mathbf{v}_c$, the calculated articulation is approximately correct. However, for non-skin-tight complex avatars, $\mathbf{p}$ may be far away from any mesh vertex, resulting in erroneous coordinate transformation causing quality issues such as extra limbs and artifacts. To avoid such problems, we further introduce a novel density weighting mechanism.

**Density weighting network.** We propose a density weighting mechanism to suppress color contribution from erroneously transformed point $\mathbf{p}$, effectively alleviating undesirable artifacts. To achieve this, we train a generalizable density weighting network $\text{MLP}_{\text{DWN}}$. More concretely, we project sampled point $\mathbf{p}$ and its closest vertex $\mathbf{v}_c$ to the canonical space via the transformation matrix $\mathbf{T}_{\text{skel}}$, embed these two coordinates with the positional embedding function $\text{PE}(\cdot)$ and then use the concatenated embeddings as inputs to $\text{MLP}_{\text{DWN}}$. The process can be defined as,

$$d' = \text{MLP}_{\text{DWN}}(\text{PE}(\mathbf{T}_{\text{skel}} \cdot \mathbf{p}) \oplus \text{PE}(\mathbf{T}_{\text{skel}} \cdot \mathbf{v}_c)). \tag{4}$$

We then compute density weights $w_d$ according to the distance $d$ between $\mathbf{p}$ and $\mathbf{v}_c$, and $d'$:

$$w_d = \text{Sigmoid}(-(d - d')/a). \tag{5}$$

where $a$ is a preset parameter. Finally, the density $\delta$ of sampled point $\mathbf{p}$ is re-weighted to $\delta \cdot w_d$ for subsequent volume rendering in NeRF.

**Sampled human pose prior.** To enable animating generated avatars with arbitrary motion sequences, we need to make sure that the density weighting network $\text{MLP}_{\text{DWN}}$ is generalizable to arbitrary poses. To achieve this, we utilize VPoser [29] as a human pose prior, which is a variational autoencoder that learns a latent representation of human pose. During training, we randomly sample SMPL pose parameters $\xi$ from VPoser to construct the corresponding posed meshes. We utilize the mesh to (1) extract skeleton maps as conditioning images for 3D-consistent SDS; (2) to serve as mesh guidance for learning animatable avatar representation. This strategy aligns avatar articulation learning with

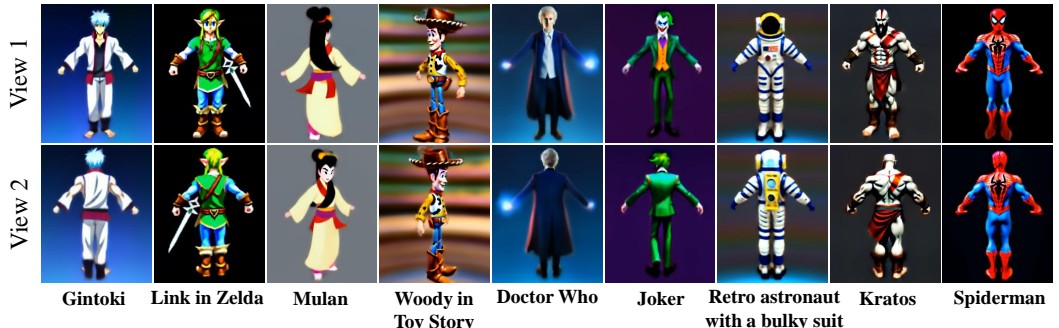

**View 1**     **View 2**

**Gintoki**   **Link in Zelda**   **Mulan**   **Woody in Toy Story**   **Doctor Who**   **Joker**   **Retro astronaut with a bulky suit**   **Kratos**   **Spiderman**

Figure 4: Qualitative results from two views of DreamWaltz. Given text prompts, it can generate high-quality 3D avatars with complex geometry and texture.

SDS supervision, ensuring that $MLP_{DWN}$ could learn a generalizable density weighting function from diverse poses. We also observe that SDS with diverse pose conditioning could further improve the visual quality of created avatar, e.g., sharper appearance.

### 3.3 Making a Scene with Animatable 3D Avatars

DreamWaltz provides an effective solution to animatable 3D avatar generation from text, readily applicable to make a scene with diverse interactions. For example, different animatable avatars could be rendered in the same scene, achieving avatar-avatar animation. In Fig. 1 (c), we can make the avatar "Woody" dance with "Stormtrooper". The animation process does not require any retraining. However, the constructed scene might exhibit unruliness, artifacts, or interpenetration issues because the naive composite rendering does not take into account the interactions between different components.

Although not our main contribution, we explore the refinement of compositional scene representation with the proposed 3D-consistent SDS. Firstly, we employ DreamWaltz and Latent-NeRF [21] to generate avatars and objects of the desired scene, respectively. Then, we manually set the motions of avatars and objects, and adopt composite rendering [13] to obtain a scene image. For scene refinement, we further provide a text prompt of the desired scene for ControlNet, and use animatable avatar learning of DreamWaltz to fine-tune the whole scene representation. Benefiting from diffusion model's scene-related image priors, the visual quality of the generated dynamic 3D scene could be further improved. More results and discussions are provided in Appendix B.4.

## 4 Experiment

We validate the effectiveness of our proposed framework for avatar generation and animation. In Sec. 4.1, we evaluate avatar generation with extensive text prompts for both qualitative comparisons and user studies. In Sec. 4.2, we demonstrate avatar animation given novel motion sequences. We present ablation analysis in Sec. 4.3 and illustrate that our framework can be further applied to make complex scenes with diverse interactions in Sec. 4.4.

**Implementation details.** DreamWaltz is implemented in PyTorch and can be trained and evaluated on a single NVIDIA 3090 GPU. For the canonical avatar creation stage, we train the avatar representation for 30,000 iterations, which takes about an hour. For the animatable avatar learning stage, the avatar representation and the introduced density weighting network are further trained for 50,000 iterations. Inference takes less than 3 seconds per rendering frame. Note that the two stages can be combined for joint training, but we decouple them for training efficiency.

### 4.1 Evaluation of Canonical Avatars

**High-quality avatar generation.** We show 3D avatars created by DreamWaltz on diverse text prompts in Fig. 4. The geometry and texture quality are consistently high across different viewpoints for different text prompts. See Appendix A.1 for more discussions and results.

**Comparison with SOTA methods.** We compare with existing SDS-based methods for complex (non-skin-tight) avatar generation. Latent-NeRF [21] and SJC [43] are general text-to-3D models. AvatarCraft [13] is not included for comparison because it cannot generate complex avatars. Furthermore, AvatarCraft utilizes a coarse-to-fine and body-face separated training to improve generation quality, which is orthogonal to our method. DreamAvatar [3] is the most relevant to our method, utilizing SMPL human prior without overly constraining avatar complexity. It is evident from Fig. 5 that our method consistently achieves higher quality in geometry and texture.

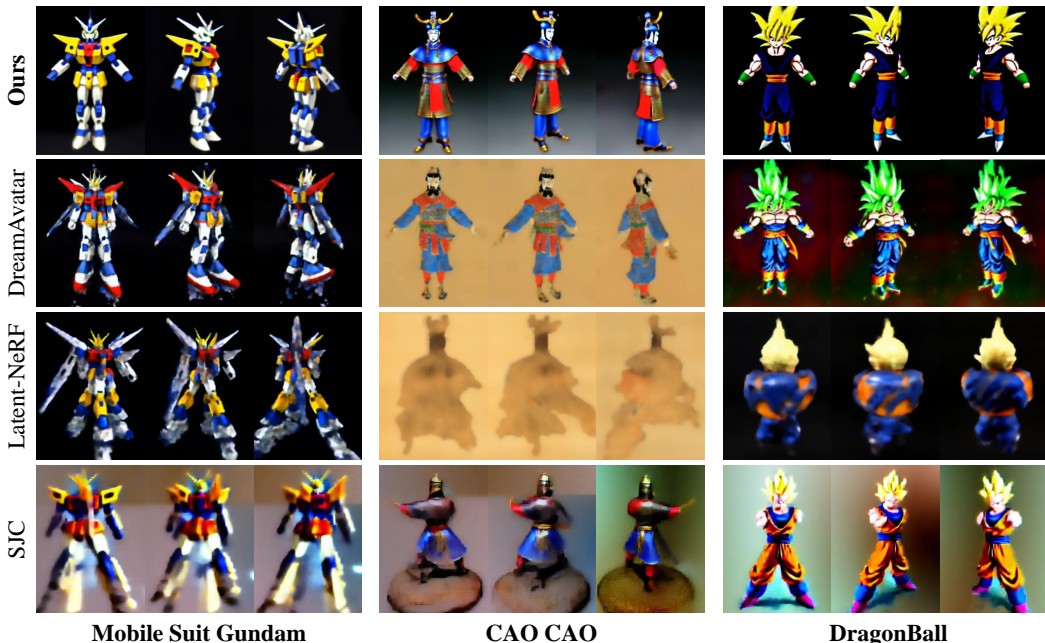

Figure 5: Qualitative comparisons for complex avatar generation. Text inputs are listed below.

**User studies.** We conduct user studies to evaluate the quality of avatar generation against existing text-to-3D generation methods: Latent-NeRF [21], SJC [43] and DreamAvatar [3]. We use the 25 text prompts DreamAvatar released and their showcase models for comparison. We asked 12 volunteers to score 1 (worst) to 4 (best) in terms of (1) avatar geometry and (2) avatar texture. We do not measure text alignment as the avatar prompts are well-known characters universally respected by all the competing models. As shown in Fig. 6, the raters favor avatars generated by DreamWaltz for both better geometry and texture quality. Specifically, DreamWaltz outperforms the best competitor DreamAvatar by score 0.67 on geometry and by score 1.01 on texture.

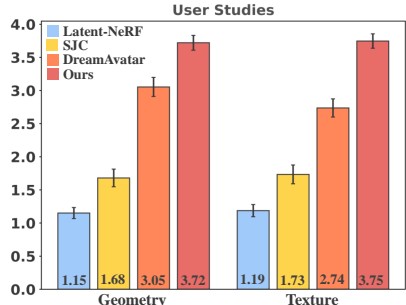

Figure 6: User preference study. DreamWaltz obtains higher scores in both geometry and texture.

### 4.2 Evaluation of Animatable Avatars

We demonstrate the efficacy of our animation learning method with animation results on two motion sequences as shown in Fig. 7: row (a) displays motion sequences in skeletons as animation inputs; our framework directly applies the motion sequences to our generated avatars "Flynn Rider" and "Woody" without any retraining. We render the corresponding normal and RGB sequences in (b) and (c). In (d), we provide free-viewpoint rendering of a chosen sequence frame. Since we essentially infer a posed 3D avatar for each motion frame, the renderings are by construction 3D-consistent and we can creatively use the 3D model for novel-view video generation.

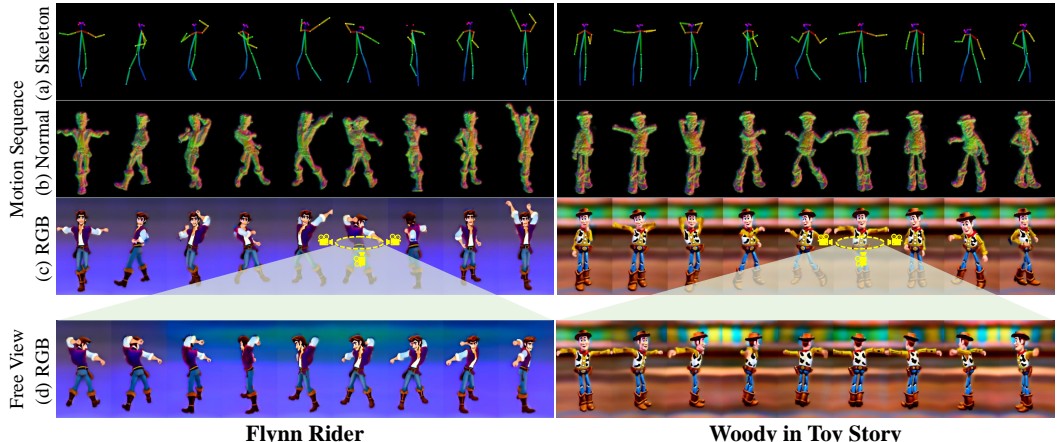

Figure 7: Animation results by applying skeleton motion sequences to animatable avatars generated by DreamWaltz. For each pose frame we could effortlessly render 3D-consistent free-viewpoints.

## 4.3 Ablation Studies

To evaluate the design choices of DreamWaltz, we ablate on the effectiveness of our proposed occlusion culling and animation learning.

**Effectiveness of occlusion culling.** Occlusion culling is crucial for resolving view ambiguity, both for 2D and 3D generation, as shown in Fig. 8 (a) and Fig. 8 (b), respectively. Limited by the view-aware capability, ControlNet fails to generate the back-view image of a character even with view-dependent text and skeleton prompts, as shown in Fig. 8 (a). The introduction of OC eliminates the ambiguity of skeleton conditions and helps ControlNet to generate correct views. Similar effects can be observed in text-to-3D generation, as shown in Fig. 8 (b).

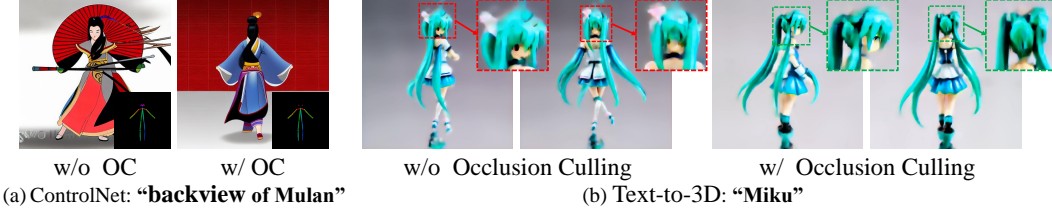

(a) ControlNet: **"backview of Mulan"**  (b) Text-to-3D: **"Miku"**

Figure 8: Ablation study on occlusion culling (OC). Occlusion culling refines the skeleton condition image by removing occluded human keypoints, which helps (a) ControlNet [48] to correctly generate character back view, (b) text-to-3D model to resolve the multi-face problem.

**Effectiveness of animation learning.** We compare our animation learning strategy with other animation methods, including: vanilla Inverse-LBS (Baseline) and AvatarCraft [13]. From Fig. 9, it is evident that our animation learning method is significantly more effective than Inverse-LBS and AvatarCraft [13]. Note that AvatarCraft cannot faithfully animate complex features, e.g. Woody's hat.

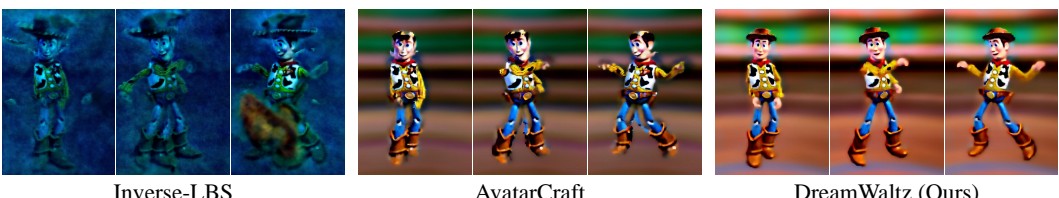

Inverse-LBS   AvatarCraft   DreamWaltz (Ours)

Figure 9: Comparison of different animation strategies, including: Inverse-LBS (Baseline), Avatar-Craft [13], and DreamWaltz (Ours). Our method learns to animate avatars from pose-conditioned image priors, therefore able to animate complex characters with significantly higher quality.

### 4.4 Further Analysis and Application

**Shape control via SMPL parameter $\beta$.** DreamWaltz allows shape adjustment for controllable 3D avatar generation. As shown in Fig. 10, the body proportions of the generated avatars can be changed by simply specifying different SMPL parameter $\beta$.

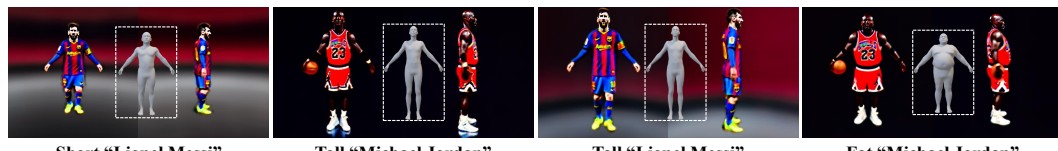

<p align="center">Short "Lionel Messi"  Tall "Michael Jordan"  Tall "Lionel Messi"  Fat "Michael Jordan"</p>

Figure 10: Our method supports adjusting the body proportions of avatars by adjusting the SMPL parameter $\beta$. For example, athlete avatars of different heights and fatness can be generated.

**Creative and diverse avatar generation.** DreamWaltz achieves stable 3D avatar generation via 3D-consistent SDS, enabling creative and diverse 3D avatar creation with low failure rates. Given imaginative text prompts and different random seeds, our method successfully generates various high-quality avatars, e.g., "a doctor wearing Woody's hat", as shown in Fig. 11.

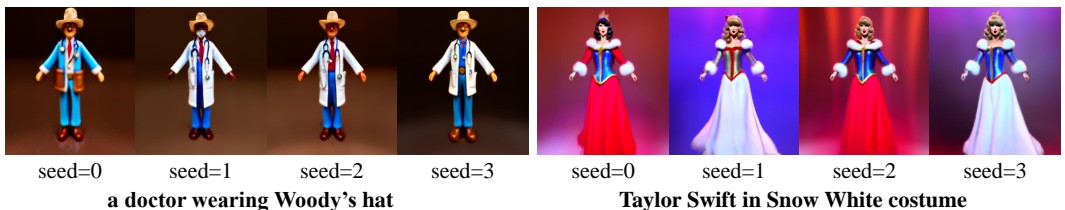

<p align="center">seed=0  seed=1  seed=2  seed=3    seed=0  seed=1  seed=2  seed=3</p>

<p align="center">a doctor wearing Woody's hat    Taylor Swift in Snow White costume</p>

Figure 11: Creative avatar creation with different random seeds and initial noises.

**Scene composition with diverse interactions.** Benefiting from DreamWaltz, we could compose animatable avatars and other 3D assets into the same scene. We give a few examples to highlight the potential for enabling diverse interactions: (1) avatar-avatar interaction as shown in Fig. 1 (c) depicting "Woody" dancing with "Stromtrooper"; and (2) avatar-scene interaction as shown in Fig. 1 (d) with "Woody" sitting on different chairs. The interactions can be freely composed to make a dynamic scene. More results are provided in Appendix A.3.

## 5 Discussion and Conclusion

**Limitation.** Although DreamWaltz can generate SOTA high-quality complex avatars from textual descriptions, the visual quality can be significantly improved with higher resolution training at higher time and computation cost. The quality of face and hand texture can be further improved through dedicated optimization of close-up views as well as adopting stronger SMPLX instead of SMPL.

**Societal Impact.** Given our utilization of Stable Diffusion (SD) as the 2D generative prior, our model could potentially inherit societal biases present within the vast and loosely curated web content harnessed for SD training. We strongly encourage the usage to be transparent, ethical, non-offensive and a conscious avoidance of generating proprietary characters.

**Conclusion.** We propose DreamWaltz, a novel learning framework for avatar creation and animation with text and human shape/pose prior. For high-quality 3D avatar creation, we propose to leverage human priors with SMPL-guided initialization, further optimized with 3D-consistent occlusion-aware Score Distillation Sampling conditioned on 3D-aware skeletons. Our method learns an animatable NeRF representation that could retarget the generated avatar to any pose without retraining, enabling realistic animation with arbitrary motion sequences. Extensive experiments show that DreamWaltz is effective and robust with state-of-the-art avatar generation and animation. Benefiting from DreamWaltz, we could unleash our imagination and make diverse scenes with avatars.

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

# Appendix

In this supplementary material, Sec. A presents more qualitative results of DreamWaltz on avatar creation, animation and interaction. Sec. B gives more analysis on the design choices of our method. Section C provides more comprehensive implementation details.

## A  More Qualitative Results

We provide more results of our proposed method, including more generated avatars in Sec. A.1, more animated avatar sequences in Sec. A.2, and more demonstrations on diverse interactions in Sec. A.3.

### A.1  Avatar Creation

We provide more text-to-3D avatar generations in Fig. 12 with a wide range of text prompts including celebrities, popular cartoon/movie characters and text descriptions. Note that DreamWaltz is capable of generating diverse avatars. For instance, we can produce avatars with a human-realistic appearance like "Tiger Woods", avatars wearing intricate clothing such as "Napoleon" and "Marie Antoinette", and avatars tailored to user-provided characteristics like "Blue fairy with wings", among others.

### A.2  Avatar Animation

We provide more animation results on six characters as shown in Fig. 13. Please refer to the project page at https://dreamwaltz3d.github.io/ for more animation sequences.

### A.3  Diverse Interaction

We provide more results of diverse interactions in Fig. 14, including: avatar-object, avatar-scene, and avatar-avatar interactions. Please refer to the videos on project page at https://dreamwaltz3d.github.io/ for the full sequences.

## B  More Analysis

### B.1  Visualization of SDS Gradients

We visualize the SDS supervision gradients $\|\boldsymbol{\epsilon}_\phi(\mathbf{z}_t; y, t) - \boldsymbol{\epsilon}\|$ for NeRF renderings in Fig. 15 (a) and the denoised images derived from noise predictions $\boldsymbol{\epsilon}_\phi(\mathbf{z}_t; y, t)$ in Fig. 15 (b). These visualizations are based on the text prompt $y$ of "superman" and $t$ of 980, while additional conditioning of depth and skeleton is provided in the second and third row, respectively, in accordance with the current rendering viewpoint of NeRF. It is evident that depth and skeleton images offer more informative optimization gradients compared to text alone. However, depth images heavily rely on the SMPL prior, leading to gradients that conform tightly to the avatar's skin, resulting in the disappearance of superman's cape. On the other hand, skeleton images as adopted by DreamWaltz provide both informative and flexible supervision, accurately capturing the avatar's shape, pose, and intricate details such as the cape.

### B.2  Effects of Random-Pose Optimization on Avatar Quality

In Fig. 16, we present visualizations of the avatars obtained at various stages, all depicted in a canonical pose. In Stage I, a static avatar is generated by optimizing its 3D representation with the canonical pose. In Stage II, the system undergoes training on randomly sampled human poses to facilitate animation learning. Although not our primary objective, this design enables the generated avatar to further refine its appearance with different poses. As a result, minor adjustments are observed, typically leading to sharper details. However, in some cases, the geometry may become more simplified, aligning more closely with the SMPL prior.

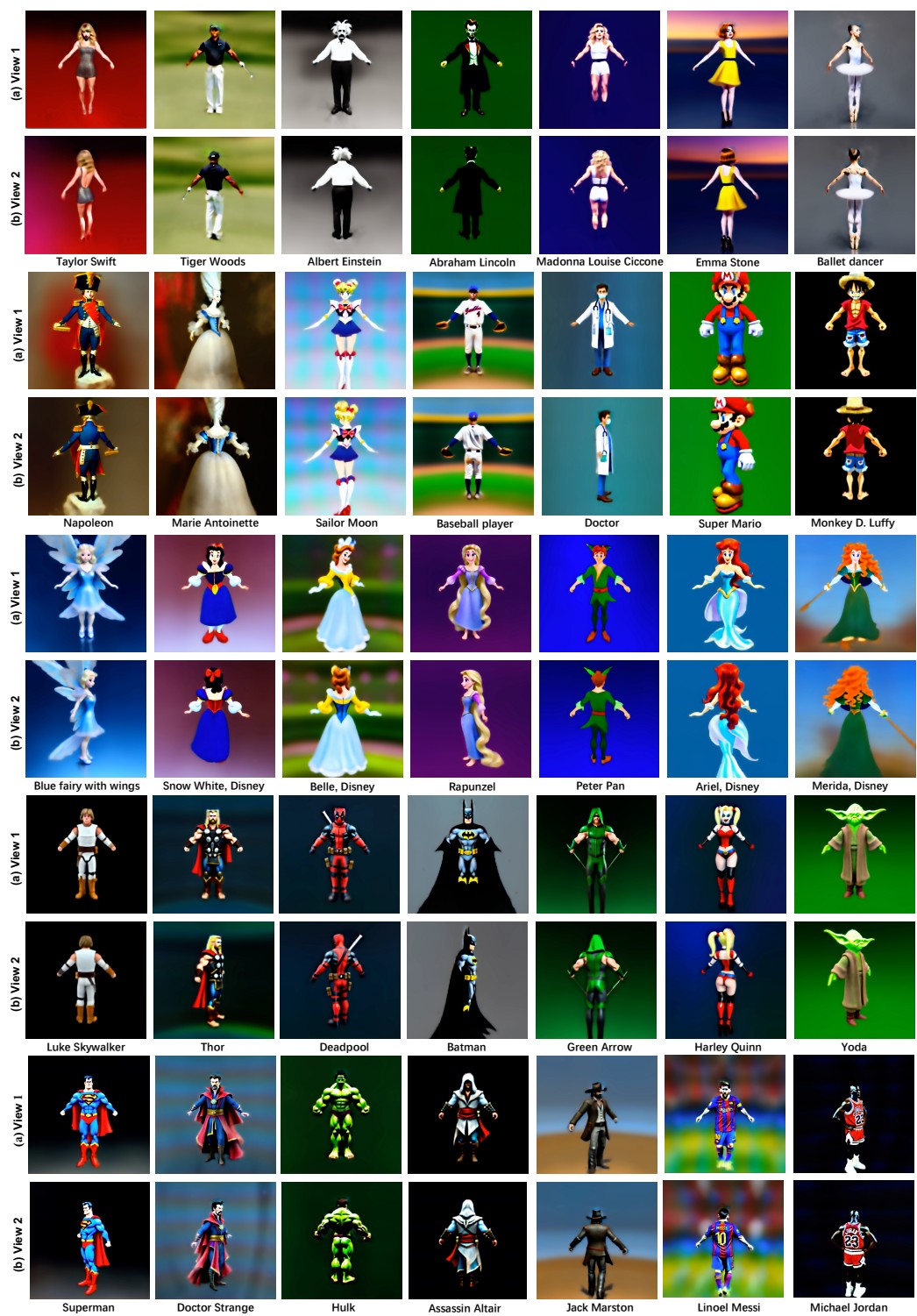

Figure 12: Text-to-3D avatars generated with DreamWaltz, each displayed for two views.

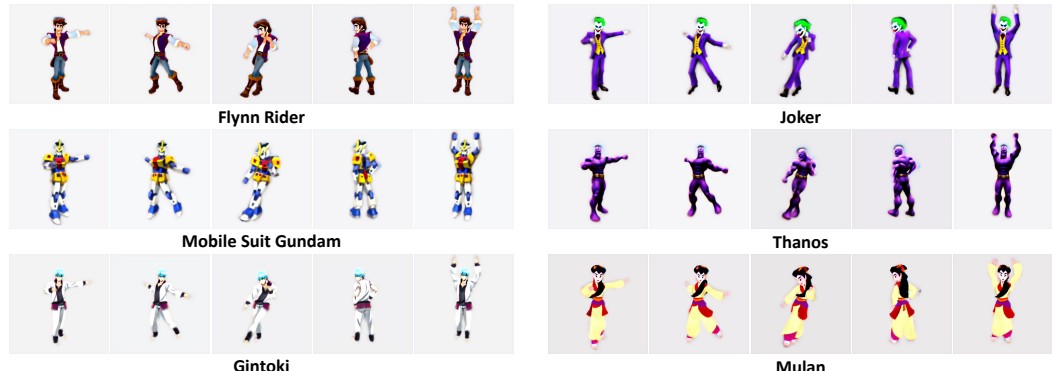

Figure 13: Avatar animations on six characters.

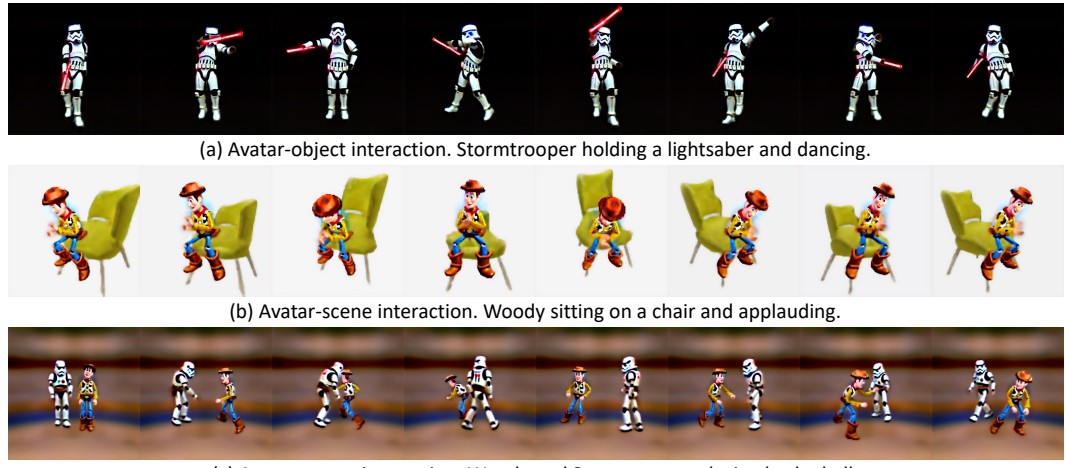

(a) Avatar-object interaction. Stormtrooper holding a lightsaber and dancing.

(b) Avatar-scene interaction. Woody sitting on a chair and applauding.

(c) Avatar-avatar interaction. Woody and Stormtrooper playing basketball.

Figure 14: Avatar animations with diverse interactions.

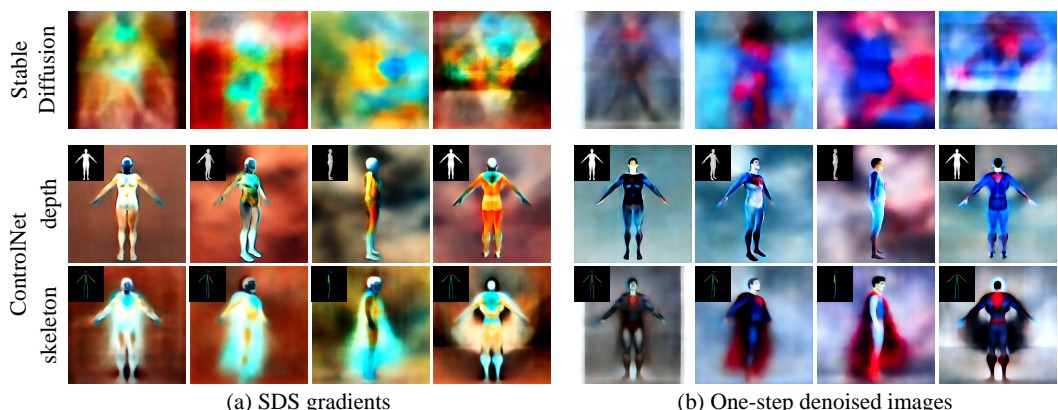

(a) SDS gradients      (b) One-step denoised images

Figure 15: Visualization of the SDS gradients (a) and the corresponding denoised images (b), given the text prompt "superman". The second and third rows are conditioned on additional depth and skeleton images, respectively, as indicated in the upper left corner of each visualization. It is clear that the skeleton image as adopted by DreamWaltz provides more informative supervision compared to text alone. Skeleton conditioning is also less restrictive than depth conditioning, successfully avoiding the disappearance of superman's cape.

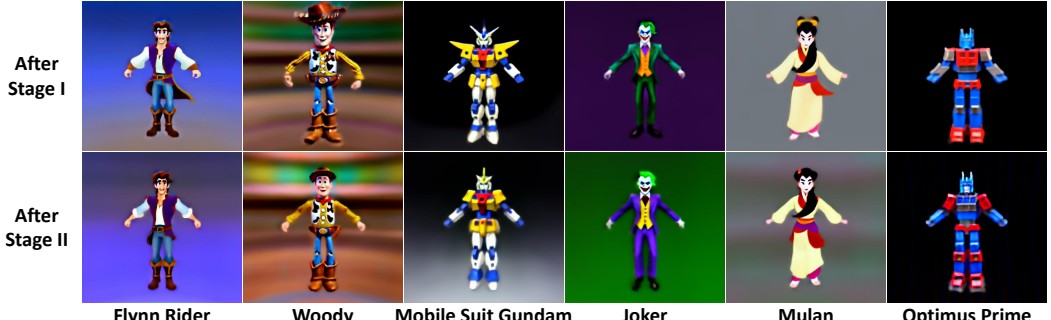

Figure 16: Visualization of canonical avatars obtained at different stages. The optimization at Stage II slightly changes the shape and appearance of avatars, resulting in sharper details (e.g., Woody's hat) but sometimes more simplified geometry (e.g., Flynn's clothes).

## B.3 Single-stage Training vs. Two-stage Training

In Fig. 17, we present a qualitative comparison between single-stage training (Stage II only) and two-stage training (Stage I + Stage II) approaches using our proposed framework, specifically for an avatar animation (e.g., on a dance motion sequence). When applied to different characters, the single-stage strategy may result in problematic body topology and noticeable artifacts. In contrast, the two-stage strategy effectively mitigates these issues, leading to improved visual quality. When employing a single-stage strategy with end-to-end training, the model is required to simultaneously learn the generation of avatar geometry and appearance, along with animation. This introduces complex optimization dynamics, leading to potential slower optimization and sub-optimal results.

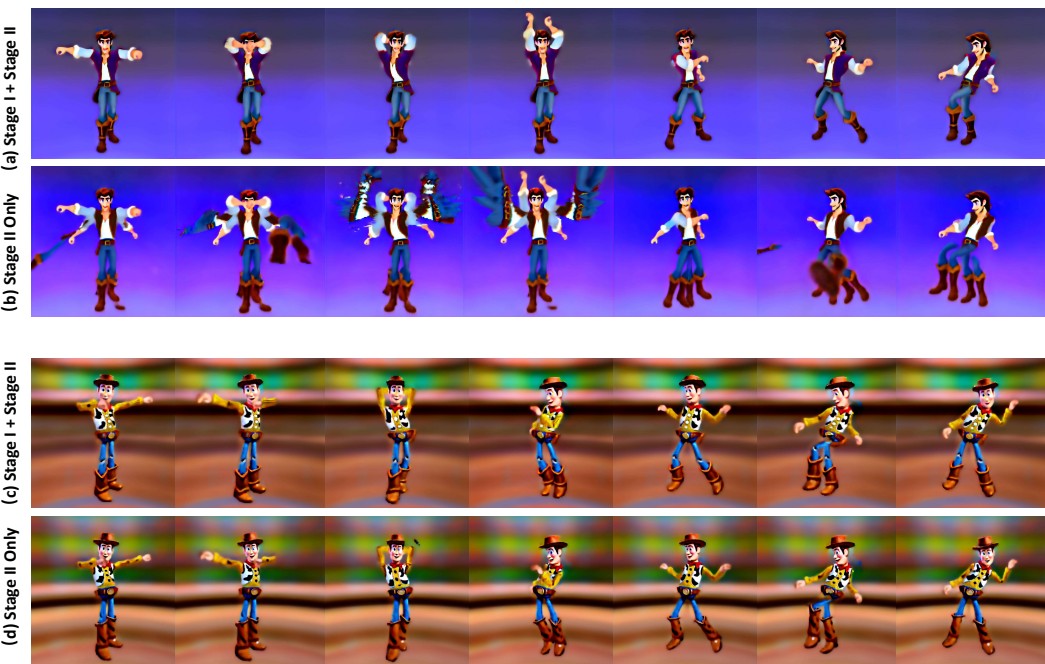

Figure 17: Qualitative comparisons of single-stage training (Stage II only) with two-stage training (Stage I + Stage II) on a dance motion sequence, both based on our proposed framework. For different characters, the single-stage strategy may suffer incorrect body topology and severe artifacts, as shown in (b). In contrast, the two-stage strategy can relieve these issues (from (b) to (a)) with better visual quality (from (d) to (c)). The one-stage strategy tends to be subjected to slow optimization speed and sub-optimal results.

## B.4 Effects of Joint Optimization for Scene Composition

Benefiting from DreamWaltz, we can create diverse animatable avatars that are prepared to engage in scenes with interactions. One approach would be to simply render different animatable avatars together in a scene. However, such composition is susceptible to issues such as artifacts and unnatural interactions. To further improve the scene quality, we could fine-tune for each specific scene. As shown in Fig. 18, fine-tuning brings noticeable improvements, such as enhancements to Woody's hat and boots, as well as more realistic "hands bumping" interactions.

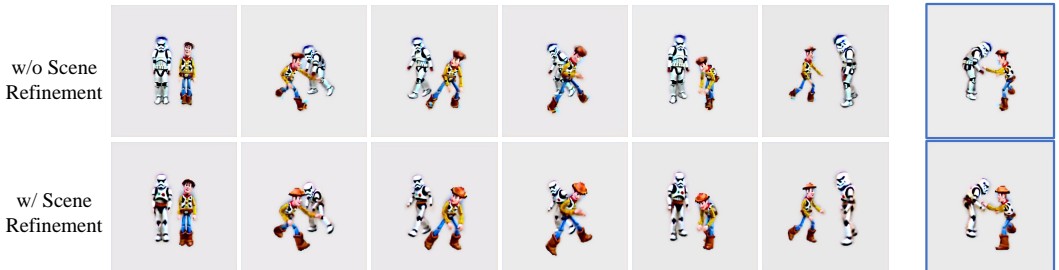

Text Prompt for Scene Refinement: "***Woody in Toy Story playing basketball with Stormtrooper***"

Figure 18: To further enhance the visual quality of complex scene generation involving multiple avatars and interactions, scene refinement (i.e. fine-tuning with our proposed 3D-consistent SDS) can be applied to eliminate artifacts. As depicted in the frames marked with blue boxes, fine-tuning brings noticeable improvements, such as enhancing the appearance of Woody's boots and achieving more realistic "hands bumping" effects.

## B.5 Realistic Animation with Pose-dependent Changes

To further demonstrate our animation performance, we provide animation results of complex character "Elsa" (with long hair and skirt) using our animation method, in comparison with animating extracted mesh with the commercial application *Mixamo*. As demonstrated in Fig. 19, with our method, Elsa's skirt and hair exhibit significantly more natural displacements and movements (as highlighted in yellow and red, respectively) as pose changes.

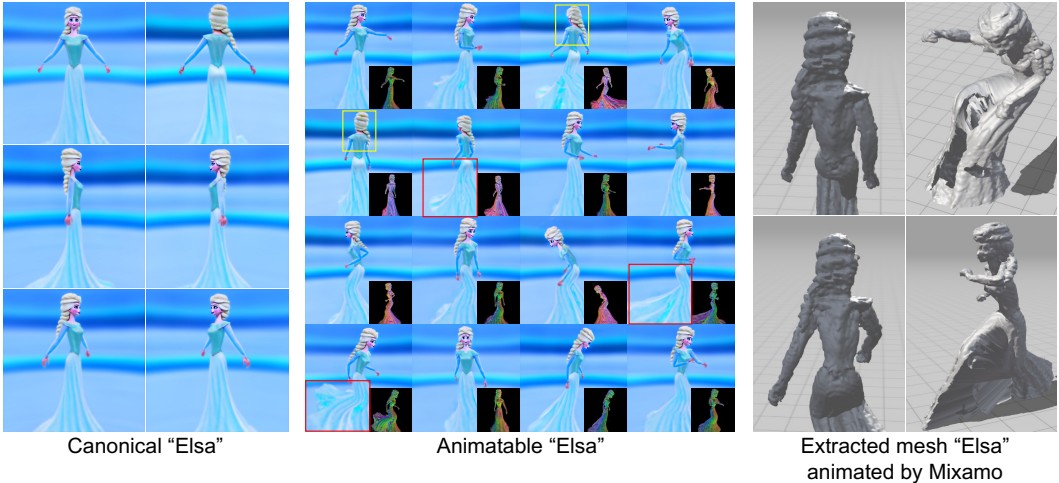

Canonical "Elsa"          Animatable "Elsa"          Extracted mesh "Elsa" animated by Mixamo

Figure 19: Animation results of complex avatar (Elsa with long hair and skirt). Our animation method could achieve high-quality realistic animation with pose-dependent changes (hair displacement and skirt movements as highlighted in yellow and red respectively), while animation with rigged mesh fails with hair stuck to left arm and unrealistic skirt.

## C  More Implementation Details

**Diffusion Guidance.** We adopt ControlNet [48] with Stable-Diffusion v1.5 [34] as the backbone to provide 2D supervision. Specifically, we utilize the score distillation sampling (SDS) technique introduced by DreamFusion [31] to obtain the back-propagation gradients of 3D avatar representation. During training, we randomly sample the timestep from a uniform distribution of $[20, 980]$, and the classifier-free guidance scale is set to $50.0$. The weight term $w(t)$ of SDS loss is set to $1.0$, and we normalize the SDS gradients to stabilize the optimization process. The conditioning scale for ControlNet is set to $1.0$ by default.

The proposed DreamWaltz utilizes two types of conditioning: text prompt and skeleton image. The text prompt is given by the user to provide avatar description. View-dependent text augmentation from DreamFusion [31] is also used:

$$
\begin{cases}
\text{``front view of...''} & \theta_{\text{cam}} \in [0°, 90°] \\
\text{``backside view of...''} & \theta_{\text{cam}} \in [180°, 270°] \\
\text{``side view of...''} & \text{otherwise,}
\end{cases}
$$

where $\theta_{\text{cam}}$ denotes the azimuthal angle of camera position. The skeleton image is exported from the 3D SMPL mesh, where the rendering view is required to be consistent with the rendering view of NeRF for training.

**NeRF Rendering.** We adopt Instant-NGP [25] as the implicit avatar representation. The ray marching acceleration based on occupancy grid is disabled for dynamic scene rendering. The 3D avatar representation renders "latent images" in the latent space of $\mathbb{R}^{64 \times 64 \times 4}$ following Latent-NeRF [21], where the "latent images" can be decoded into RGB images of $\mathbb{R}^{512 \times 512 \times 3}$ by the VAE decoder of Stable Diffusion [34]. During training, the camera positions are randomly sampled in spherical coordinates, where the radius, azimuthal angle, and polar angle of camera position are sampled from $[1.0, 2.0]$, $[0, 360]$ and $[60, 120]$, respectively.

**Optimization.** Throughout the entire training process, we use Adam [16] optimizer with a learning rate of 1e-3, and batch size is set to 1. For the canonical avatar creation stage, we train the avatar representation for 30,000 iterations, which takes about an hour on a single NVIDIA 3090 GPU. For the animatable avatar learning stage, the avatar representation and the introduced density weighting network are further trained for 50,000 iterations. Inference takes less than 3 seconds per rendering frame. We further fine-tune the hybrid avatar representations for 30,000 more steps for scenarios with multiple avatars and complex interactions.

**Dataset.** To create animation demonstrations, we utilize SMPL-format motion sequences from the 3DPW [42] and AIST++ [17] datasets to animate avatars. SMPL-format motion sequences extracted from in-the-wild videos are also used.

