# OpenReview forum: "DreamWaltz: Make a Scene with Complex 3D Animatable Avatars"
_NeurIPS.cc/2023/Conference — NeurIPS 2023 poster_

### Official Review · Reviewer_rngx · 2023-07-02

**Soundness:** 3 good
**Presentation:** 3 good
**Contribution:** 2 fair
**Rating:** 4
**Confidence:** 5

**Summary:**

The paper presents an approach for creating animatable avatars from text prompts. It builds on DreamFusion and makes it articulated by incorporating articulated NeRF and SMPL body model. It also replaces the vanilla text-to-image model (StableDiffusion) with ControlNet to introduce 3D consistent SDS loss. The performance is evaluated using a user study where the proposed method is shown to outperform existing methods.

**Strengths:**

- The paper addresses the challenging problem of creating neural avatars from text prompts.
- The paper demonstrates that DreamFusion can be extended to articulated humans.
- The proposed method is intuitive and makes sense, though it is not entirely novel.

**Weaknesses:**

### Novelty
- The main limitation of the paper is the lack of novelty. The paper builds on DreamFusion and then adopts existing methods for articulated NeRF. It replaces the vanilla stableDiffussion model with ControlNet for 3D consistent SDS loss. Overall, the proposed approach is a straightforward combination of existing methods and it is hard for me to identify any novel technical contribution of the paper.

### Baselines
- A simple baseline is missing. Similar to AvatarClip, we can extract a static mesh using MarchingCube and then rig it with SMPL. This will make it animatable. Since the paper's main contribution is animatable avatars, I believe having this baseline is important to properly validate the contributions of the paper.

### Visual quality
- The quality of the generated avatars is also limited even though they are all well-known characters.



**Questions:**

### Implementation Details:
- Would be nice to provide further implementation details. What is the batch size, learning rate, guidance value, etc.?
- How is the background learned during training, what are light settings, etc?

### Ablation Studies:
- What is the impact of the density-weighted network? Existing methods for neural avatars (AniSDF, HumanNeRF, etc.) use different variants to accommodate non-rigid deformations. How does the performance compare with those?
- What happens if random body poses from humans prior are not used during training? Would the DWN module still work? Since it operates in the canonical space, why random body poses are required in the first place?

**Limitations:**

The paper does not discuss the limitations of the proposed approach. Overall, the visual quality of the generated avatars is quite limited and there are severe artifacts in the animations (e.g., on the arms of Woody in support). The time required to generate an avatar is also a limitation of the proposed method and should be discussed. The qualitative results are mostly provided for the cartoonish characters which are well known. It would have been nice to see more creative avatars e.g., a doctor with a Woody's Hat, etc.

---

> ### Author Rebuttal · Authors · 2023-08-10
>
> Thank you for the detailed feedback and questions! Below we address the questions and concerns separately.
>
> ### **Q: The paper is a straightforward combination of existing methods and lacks novelty.**
>
> **A:** Our work addresses two important problems in text-dirven avatar generation:
>
> **i)** SDS-based methods struggle to provide view-consistent supervision for avatar creation. Our work proposes 3D-consistent SDS with occlusion culling by utilizing SMPL and ControlNet, effectively increasing generation quality and avoiding multiple faces.
>
> **ii)** Images contain rich information of human poses and interactions. We distill this information from diffusion model to avatar through arbitrary pose conditioned SDS, reducing the dependence on SMPL and allowing avatars with complex shapes to be animated.
>
> Besides, compared to existing text-to-avatar works AvatarCLIP, AvatarCraft, and DreamAvatar, our work achieves the best visual quality and for the first time allows animation of complex avatars.
>
> ### **Q: A mesh-based animation baseline is missing.**
>
> **A:** The solution based on mesh extraction and SMPL animation have two disadvantages.
>
> **i)** There is a serious loss of fidelity in the conversion from the implicit field to mesh.
>
> **ii)** SMPL only describes the naked human body and is not suitable for animating complex shapes, which is why the avatars created by AvatarCLIP are all SMPL-like shapes.
>
> For comparison, we further provide animated results by mesh-based method and ours in Fig. 5 of the rebuttal pdf. Even with the professional tool Mixamo (requires manual rigging), the mesh-based animated results are still inferior to ours.
>
> ### **Q: The quality of the generated avatars is limited.**
>
> **A:** Our results already outperform existing text-driven sota works with much faster generation speed (46% and 50% training time of AvatarCraft and DreamAvatar). Both reviewer kXf3 and DdUA gave positive comments on our experimental results, such as: “Experiments show SOTA performance for text-driven avatar generation and animation,” “The comparison with Sota is fair and complete to the best of my knowledge. The generated static avatars are in good quality.” The quality could be further improved via mesh refinement or zoom-in training but won’t be our contribution.
>
> ### **Q: Would be nice to provide further implementation details.**
>
> **A:** We gently remind the reviewer that these implementation details are already provided in Sec. 3 of the supplementary material. Throughout the entire training process, we set the batch size to 1, set the classifier-free guidance scale to 50, and use the AdamW optimizer with a learning rate of 1e-3.
>
> ### **Q: How is the background learned during training, what are light settings?**
>
> **A:** We use a two-layer MLP network as background NeRF to model the background which is common in DreamFusion implementations. We encode the ray direction using frequency encoding and utilize it as an input to the background NeRF, which predicts background color solely based on the ray direction. During training, the background NeRF is also optimized by SDS gradients, with the learning rate of 1e-3.
>
> The lighting modeling remains as future work (also not yet considered in related work of DreamAvatar and AvatarCraft) since our work focuses on text-to-avatar generation.
>
> ### **Q: What is the impact of the density-weighted network?**
>
> **A:** The density-weighted network mitigates animation artifacts due to inaccurate warping by setting the density of the points far from the target mesh surface to zero.
>
> ### **Q: How is the performance of the density-weighted network compared to different variants for non-rigid deformations?**
>
> **A:** We did try various modules for non-rigid deformations, such as the non-rigid motion module from HumanNeRF, but found that the network failed to learn the correct deformations (severe artifacts appeared). We think the reason is that learning deformations from arbitrary pose-conditioned ControlNet is much more difficult than from videos.
>
> ### **Q: Why are random body poses required during training? Would the DWN module still work without random body pose?**
>
> **A:** The DWN module will not work if random body poses are not used during training. The inputs of DWN involve the ray sampling point $p$ and its nearest neighbor vertex $v_c$, while the nearest vertex $v_c$ is related to current body pose. If the body pose is fixed, DWN will degenerate to a constant value, similar to the animation method of AvatarCraft which won’t be able to animate complex avatars.
>
> ### **Q: The paper has limitations in visual quality, time cost, and avatar creativity.**
>
> **A:** Although not fully satisfactory, our method outperforms previous text-driven avatar creation works in terms of visual quality, time cost, and avatar creativity.
>
> **a) Visual quality.** For static avatar creation, our results achieve sota visual quality compared to previous works on text-driven avatar creation. For animation, the severe artifacts of the Woody case are actually not common, and we provide more animation results in Fig. 2 and Fig. 5 of the rebuttal pdf.
>
> **b) Time cost.** To generate a canonical avatar, our work takes about one hour on one NVIDIA A100 GPU, significantly outperforming existing text-driven avatar creation works. For comparison, AvatarCLIP takes 5 hours on one NVIDIA A100 GPU, AvatarCraft takes 2.2 hours on one NVIDIA A100 GPU, DreamAvatar takes about 2 hours on one NVIDIA RTX 2080Ti GPU.
>
> **c) Creative characters.** We mainly provide cartoon characters for a fair comparison with previous text-driven avatar creation works. Many photorealistic avatars of celebrities are provided in Fig. 1 of the supplementary material pdf, including “Taylor Swift”, “Albert Einstein”, “Emma Stone”, “Linoel Messi”, etc. Following the reviewer's suggestion, we provide more creative avatars in Fig. 4 of the rebuttal pdf, such as “a doctor with Woody's Hat” and “Taylor Swift in Snow White costume”.

---

### Official Review · Reviewer_DdUA · 2023-07-02

**Soundness:** 2 fair
**Presentation:** 3 good
**Contribution:** 2 fair
**Rating:** 4
**Confidence:** 4

**Summary:**

This paper proposes a new method for text-to-3D avatar generation. The proposed pipeline has two stages. The first stage generates a static avatar while the second stage learns the deformation properties of the avatar for animation. The authors propose 3D-consistent occlusion-aware score distillation sampling which seems to improve the generation quality over previous methods with standard score distillation sampling. The paper further includes the results of animation and interaction of the generated avatars.

**Strengths:**

**Method:** The proposed 3D-consistent Score Distillation Sampling and occlusion culling is reasonable and technically sound. They seem to be effective in the provided ablation study.

**Experiment:** The comparison with Sota is fair and complete to the best of my knowledge. The generated static avatars are in good quality.

**Presentation:**. The paper is well-structured and written. I find it easy to follow and understand.

**Weaknesses:**

1. I have some doubts regarding the relationship to the existing work in Table 1. DreamAvatar is marked as non-animatable, however, the original DreamAvatar paper does show reposing results in Fig 4 in their paper. This seems to contradict the claim in this paper.  And regarding the interaction between avatars/objects, shouldn't this be possible with all methods as it is basically composited volume rendering if I understand correctly?

3. In L34, the authors mention that realistic animation involves changing texture and shape in different poses. However, in the qualitative results in the videos, I did not see such pose-dependent changes but only articulation with LBS. It would be helpful to visualize the pose-dependent changes in the canonical space to understand if the model indeed learns the pose-dependent effects.

4. In video 00:00-00:01, on the leftmost sequence, Woody’s upper legs disappear when they are crossing. Why do such artifacts happen?

I am on the negative side currently. The main contribution of this paper w.r.t previous works seems to be making 3D text-to-avatars "animatable" and "interactive" for the first time (L49, table 1). However, regarding "animatable", previous work DreamAvatar does demonstrate avatars in new poses, and it's unclear why DreamAvatar is not "animatable". Also, the animation quality in this paper is still not satisfactory - there are artifacts (legs disappearing) and no obvious pose-dependent effects. Regarding "interactive", it seems that this is achieved simply by rendering two avatars in a composite way, which is straightforward and can be done with all previous methods. I hope the authors to clarify the contributions of this paper w.r.t previous methods.

**Questions:**

1. Why is DreamAvatar not regarded as non-animatable?

2. How does the learned pose-dependent shape and texture change look in the canonical space?

3. What causes the disappearing leg artifacts in the video?

**Limitations:**

There is no limitation discussed in the paper.

---

> ### Author Rebuttal · Authors · 2023-08-10
>
> Thank you for the thoughtful feedback and valuable questions! Below we address the questions and concerns separately.
>
> ### **Q: Why is DreamAvatar regarded as non-animatable?**
>
> **A:** Although DreamAvatar can repose the avatar via **retraining**, it is impractical for animation due to the inefficiency (it takes two hours for a novel pose) and inconsistency (the appearance of the avatar in different poses may be changed, as shown in  Fig. 1 of the rebuttal pdf). In fact, DreamAvatar doesn't claim to be able to animate avatars, nor does it provide animation results. For the above reasons, we believe DreamAvatar can be safely marked as not animatable.
>
> ### **Q: Interaction between avatars/objects should be possible with all methods as it is basically composited volume rendering.**
>
> **A:** Regarding the interaction between avatars/objects, yes, naive interactions should be possible for all methods via composited volume rendering, and we will modify some ambiguous expressions such as Table 1 and Line 262. Different from existing works, we explore the 2D image prior (where social interactions are abundant) from diffusion model for improving scene interaction. Specifically, we find that scene-specific fine-tuning with our proposed 3D consistent SDS can further eliminate artifacts and make interactions more realistic, as shown in Fig. 5 in the supplementary material.
>
> ### **Q: How does the learned pose-dependent shape and texture change look in the canonical space?**
>
> **A:** We provide a visualization of pose-dependent changes in Fig. 5 of the rebuttal pdf, where character Elsa's skirt and hair are changed with pose changes. Unfortunately, these changes cannot be displayed in canonical space because our animation operations are irreversible.
>
> ### **Q: What causes the disappearing leg artifacts in the video?**
>
> **A:** The disappearing leg artifacts are caused by wrong predictions of the density weighting network, which makes the masking area around the legs part too large. Due to the instability of SDS optimization, the density weighting network is difficult to converge to an optimal state for the Woody case. But this situation is not common, most characters' upper legs don't disappear when they are crossing: e.g. Fig. 1 (a) of the main paper, Fig. 2 of the supplementary material.
>
> ### **Q: Contributions of this paper w.r.t previous methods.**
>
> **A:** The main contribution of this paper is making text-driven 3D avatars “complex” and “animatable” for the first time. Previous text-to-avatar works AvatarCLIP and AvatarCraft are animatable, but the avatars generated by them are required to be highly similar to SMPL templates and oversimplified in shape. DreamAvatar is not animatable because it requires retraining for each pose control. For animation, despite rare artifacts, our method can animate various complex avatars without manual rigging and pose-dependent effects are provided in rebuttal pdf Fig. 5. For making scenes with interactions, our work not only renders two avatars in a composite way, but also proposes a scene-specific fine-tuning with our proposed 3D-consistent SDS.
>
> ### **Q: Missing Limitations.**
>
> **A:** We have supplemented the discussion of limitations in global response.

---

> > ### Comment · Reviewer_DdUA · 2023-08-15
> >
> > Thank the authors for the clarifications. However, I found the results still not strong enough to claim animatable and complex as the main contribution - the shape and animation quality are not yet satisfactory, and the artifacts in animation seem to be inherent to the method due to the instability.

---

> > > ### Author Response · Authors · 2023-08-15
> > >
> > > Thanks for the reviewer's valuable comment. Our results for complex static avatars are sota with better quality and shorter generation time than previous text-to-avatar works. For animation, although current SDS-based methods have inherent instability, using rich 2D image priors to learn 3D avatar animation is novel and worth exploring, avoiding that inverse LBS cannot be generalized to complex avatars. The Elsa example in the rebuttal pdf demonstrates pose-dependent changes learned from image priors, and we hope these results encourage more exploration of complex avatar animation.

---

### Official Review · Reviewer_1xa2 · 2023-07-04

**Soundness:** 3 good
**Presentation:** 2 fair
**Contribution:** 3 good
**Rating:** 5
**Confidence:** 4

**Summary:**

The work proposes a method for generating 3D skeleton animatable characters by distilling a latent diffusion model. It uses Control Net to add additional key point map conditioning to the diffusion process, improving the granularity of pose control. It uses DreamFusion to distill a NeRF model given a text prompt and the skeleton conditioning in a fixed A-pose. The model is then refined by sampling additional poses from the SIMPL model used as a skeleton prior. The model is shown to be able to generate animatable models from diverse text prompts that does not require re-training for new novel poses.

**Strengths:**

The method proposes a stable training regime, starting with retraining using a SIMPLE mask, followed by single pose and fine-tuning on multiples poses. The addition of depth culling also helps remove artifacts seen in prior methods.

**Weaknesses:**

- The work's main claim of animatability needs further evidence. The results show animation only for non-complex characters; no characters with long skirts or hair are animated, only reconstructed. For the samples that are shown, the animation quality is not a big departure from LBS deformation. Furthermore, there is a lack of diversity in aspects of pose that are not included in the skeleton model, for example hands are blurring and biased to a specific pose.
- The ability of the method to capture interactions appears limited to blending in 3D, where complex interactions between parts are not modeled, presumably due to the independent training. For example, the hands are not clasped in the Waltz Fig. 3
- The generated models exhibit unnatural body proportions due to the mismatch between the skeleton conditioning and the text conditioning. For example, Michael Jordan (a basketball player) and Lionel Messi (a football player) seem to have the same body proportions in Fig 1. of the Supplementary.


**Questions:**

- L22. what is the resolution of the model for rendering at 3s.
- L228. Does joint training affect generalization performance? A comparison would be useful here.
- How consistent is quality of results? Results showing reconstructions using different initial noise values would demonstrate diversity of generations.
- L272. Is inverse LBS just the proposed method without the additional MLP for d'? Clarification would be useful.
- Eqn (5). The motivation of this formula is not apparent from the text. Why would Sigmoid(d') not work similarly?

**Limitations:**

Authors have not addressed limitations or societal impact.
It is suggested that the authors outline weaknesses mentioned above and suggest remediation strategies as future work.
Social impact follows prior works such as DreamFusion.

---

> ### Author Rebuttal · Authors · 2023-08-10
>
> The authors are grateful for the reviewer's valuable feedback and insightful questions. We are encouraged by your support for this work! Below we address the concerns separately.
>
> ### **Q: Further evidence of animatability is needed. No animation results of complex characters (with long skirts or hair) are shown.**
>
> **A:** We show an animation result of Mulan (long hair) in Fig. 2 of the supplementary material. We further provide more animation results in Fig. 5 of the rebuttal pdf, animating Elsa with a long skirt and hair. As shown in Fig. 7(c) of our paper, our learnable animation method is significantly more effective than baseline methods Inverse-LBS and AvatarCraft. Intuitively, it makes sense that our animation method learned from diffusion image priors is better than the baseline methods that only rely on nearest neighbor vertex query and SMPL’s inverse-LBS, because SMPL only describes the naked human body and is not suitable for complex shapes.
>
> ### **Q: Lack of pose diversity in the skeleton model.**
>
> **A:** It is true that the pose of the skeleton model lacks diversity and may cause blurring and artifacts. But by upgrading the skeleton model to SMPLX and ControlNet to v1.1, these shortcomings can be alleviated.
>
> ### **Q: The ability of the method to capture interactions appears limited to blending in 3D.**
>
> **A:** In addition to independent training and blending in 3D, we also explore scene-specific fine-tuning, which puts multiple avatars in the same scene for joint training with the proposed 3D-consistent SDS loss (where image priors would “correct” unrealistic interactions), demonstrating improvements in visual quality as shown in Fig. 5 of the supplementary material pdf.
>
> ### **Q: Unnatural body proportions.**
>
> **A:** The body proportions can be freely controlled via specifying SMPL model parameters. We provide results of using different SMPL shape parameters to control body proportions in Fig. 3 of the rebuttal pdf.
>
> ### **Q: What is the resolution of the model for rendering at 3s.**
>
> **A:** For animation rendering, our 3D avatar representation renders 128x128 latents. Then, the latents are decoded by the VAE decoder of Stable-Diffusion to obtain 1024x1024 rgb images. The whole rendering and decoding process takes less than 3 seconds. The speed bottleneck is mainly in nearest neighbor vertex queries of ray sampling points for inverse-LBS. We use CPUs for these computations due to the huge memory requirement. Improving rendering speed remains for future work where we could consider writing faster GPU operators and more efficient nearest neighbor querying.
>
> ### **Q: Does joint training affect generalization performance?**
>
> **A:** In fact, we provide a comparison of two-stage training (Stage I + II) and joint training (Stage II only) in Fig. 7 of the supplementary material pdf. The results show that joint training does hurt generalization performance, leading to more artifacts and “phantom limbs”. More discussions are given in sec 2.4 of the supplementary material.
>
> ### **Q: How consistent is the quality of results when using different initial noise values?**
>
> **A:** We provide these results in Fig. 4 of the rebuttal pdf. The quality of avatars with different initial noises are consistently good.
>
> ### **Q: Is inverse LBS in L272 just the proposed method without the additional MLP for d'?**
>
> **A:** Inverse-LBS in L272 is the proposed method without additional MLP for d’ and NeRF fine-tuning in Stage II. We will add this clarification in the revision.
>
> ### **Q: The motivation of Eqn (5) is missing. Why would Sigmoid(d') not work similarly?**
>
> **A:** Sorry for the missing motivation of Eqn (5). We derive this formula from the mask function ${\eta(\boldsymbol{p})}$ in the AvatarCraft paper. This function aims to set the density of the points far from the target mesh surface to zero, formulated as:
>
> $$
> \eta(\boldsymbol{p})=
> \begin{cases}
> 0, \text{if } d(\boldsymbol{p}) > \delta,\\\\
> 1, \text{if } d(\boldsymbol{p}) \le \delta,
> \end{cases}
> $$
>
> where $d(\cdot)$ is the distance function between the ray sampling point $\boldsymbol{p}$ and its nearest neighbor vertex, $\delta$ is a constant threshold. Such a constant threshold is not suitable for avatars with complex shapes, so our work use the MLP-predicted value $d’$ (i.e., Eqn (4) in our paper) to replace the threshold $\delta$, resulting in:
>
> $$
> w_d:=\eta(\boldsymbol{p})=
> \begin{cases}
> 0, \text{if } d(\boldsymbol{p}) - d' > 0,\\\\
> 1, \text{if } d(\boldsymbol{p}) - d' \le 0.
> \end{cases}
> $$
>
> Finally, we introduce a sigmoid function with preset parameter $a$ to make it smooth and differentiable, obtaining Eqn (5):
>
> $$ w_d = \text{Sigmoid}(-(d-d')/a). $$
>
> An implementation of $ \text{Sigmoid}(d') $ might also work similarly but lacks intuition.
>
> ### **Q: Missing Limitations and Societal Impacts.**
>
> **A:** We thank the reviewer for the valuable suggestions. We have provided the discussion of limitations and societal impacts in the global response.

---

> > ### Comment · Reviewer_1xa2 · 2023-08-18
> >
> > Thank you for the rebuttal and additional results.
> >
> > Most of my questions are answered. I am still leaning positive, but the quality improvements using the proposed approach is not large enough to increase my rating further.
> >
> > > The finetuning of interactions does appear to improve results, however, it seems to do so mainly with regards to aspects that are independent of the interactions (e.g. the boots in Fig 5 or appendix, the "hand bumping" claim is over stated).
> > > Improvements that may come from more expressive mesh conditioning is only speculated.
> > > The body proportions follow the conditioning mesh and the rebuttal shows that it can be controlled. However, one would expect to distill such information from the model. A method that doesn't rely on the mesh conditioning may do this better. The current framework would require a separate system for estimating body proportions, which may not be trivial for non-realistic human characters.

---

> > > ### Author Response · Authors · 2023-08-18
> > >
> > > Thank you for the valuable feedbacks and support!
> > >
> > > **Interactions.** Due to the use of far camera views when finetuning scenes with interactions, the quality improvements shown in the current manuscript are mainly in interaction-independent aspects like boots. More convincing results can be obtained by focusing camera views on areas of interactions (e.g., holding hands) while fine-tuning.
> > >
> > > **Body proportion control.** The current framework still needs to manually tune the shape parameters of SMPL for body proportion control. But considering that SMPL and mesh rendering can be differentiable, it is feasible to automatically adjust shape parameters with SDS gradients, which remains as future work.

---

### Official Review · Reviewer_kXf3 · 2023-07-05

**Soundness:** 3 good
**Presentation:** 2 fair
**Contribution:** 3 good
**Rating:** 4
**Confidence:** 5

**Summary:**

This work proposes a method for text-driven human avatar generation. It combines animatable human nerf and diffusion model to implement avatar generation and animation. Extensive experiments demonstrate that its performance outperforms existing works. Also, this work supports avatar-avatar, avatar-object, and avatar-scene interactions.

**Strengths:**

1. This is among the first diffusion-based works that can generate animatable 3D avatars, which also support the interactions between avatars and scenes/objects.
2. The occlusion culling method is well-motivated and it can address the multi-face issue effectively together with the carefully selected text prompt for viewpoints.
3. Experiments show SOTA performance for text-driven avatar generation and animation.

**Weaknesses:**

1. This work is a little bit overclaimed (line 49-51) because there exist several prior works which can generate animatable 3D avatars with complex shapes and appearances, such as [1,2].
2. The authors claim that DreamWaltz is able to make a scene with diverse interactions across avatars, objects, and scenes. However, it is difficult for me to evaluate whether this point is technically challenging because there is no specific design or module to enable these interactions in the proposed framework. Please explain more about this point because it is the main contribution of this work as it is mentioned in the title.
3. The proposed framework is similar to AvatarCraft, the only difference is the SMPL inverse skinning and ControlNet.Please explain more about the differences between DreamWaltz and AvatarCraft.
4. It is confusing to classify AvatarCraft as not animatable (Table 1) because AvatarCraft can also deform the canonical avatar to different target poses in the observation space. I believe by using SMPL, DreamWaltz can only change the pose parameters to implement avatar animation, which is quite similar to AvatarCraft.
5. The authors might use the term 'generalizable NeRF' carefully because when we use 'generalizable NeRF', it usually refers to the NeRF model that is not overfitted to one 3D scene but can generalize to any inputs instead. It is highly recommended to think about this term again to avoid confusion or ambiguity because the NeRF model which can be deformed to any target pose is usually named as deformable/animatable instead of generalizable.
6. I think this work is technically solid and has good performance, but there exists large improvement room for the current manuscript.

[1] EVA3D: Compositional 3D Human Generation from 2D Image Collections.
[2] Avatargen: a 3d generative model for animatable human avatars.


**Questions:**

How to implement avatar-object and avatar-scene interaction? Please introduce more details for these two applications.

**Limitations:**

Not addressed. There is no discussion of broader societal impacts.

---

> ### Author Rebuttal · Authors · 2023-08-10
>
> The authors are grateful for the detailed and in-depth feedback from the Reviewer. We have substantially revised the manuscript as suggested by the reviewer. Below we address the mentioned concerns separately.
>
> ### **Q: Overclaiming in Line 49-51: “for the first time capable of generating avatars with complex shapes and appearance.”**
>
> **A:** We apologize for the misleading statement in Line 49-51: “for the first time capable of generating avatars with complex shapes and appearance, ready for high-quality animation and interaction”. Our work focuses on the challenging text-to-avatar generation using pre-trained vision-language models, different from EVA3D and AvatarGen which targets a specific avatar domain using fashion images. Existing works of AvatarCLIP, AvatarCraft, and DreamAvatar are more comparable. Compared to these works, our work can create both complex (non-SMPL-like) and animatable avatars from text, which is why we claim "for the first time". We appreciate the reviewer's comment and will revise the statement in Line 49-51.
>
> ### **Q: More explanation about making a scene with diverse interactions.**
>
> **A:** Making a scene with diverse interactions is challenging because it requires a deep understanding of the intricate interplay between body poses, objects, shape, and proximity. These interactions are hard to model by hand, and a recent work [1] later than ours proposes to learn human interaction priors from large image collections. Our work similarly employs image priors (where social interactions are abundant) from a pretrained diffusion model to enable more realistic scene interactions. Specifically, we use the proposed 3D-consistent SDS for composite scene NeRF fine-tuning. The scene-specific SDS gradients from the diffusion model can enhance the visual quality of the scene involving avatars and interactions, for example, make “hands bumping” effects more realistic, as shown in Fig. 5 of the supplementary material.
>
> [1] Generative Proxemics: A Prior for 3D Social Interaction from Images. Arxiv 2023.
>
> ### **Q: Differences between DreamWaltz and AvatarCraft.**
>
> **A:** AvatarCraft and our DreamWaltz are both two-stage approaches, where a canonical avatar is first created and then animated, but the details and results are significantly different.
>
> The biggest disagreement is whether animation is learnable. AvatarCraft only borrows the inverse-LBS from SMPL to animate the implicit field, with a hard-coded mask function to filter out the points far from the template mesh. Also based on inverse-LBS, our DreamWaltz allows the implicit field and mask function to be further learned under the supervision of rich image prior: the controlnet supervision from any pose condition. This improvement is not trivial, as it gets rid of the over-reliance on the SMPL mesh topology, allowing us to animate avatars with more complex (non-SMPL) shapes (as shown in Fig. 7(c) of our paper), which is crucial for imaginative text-driven avatar creation.
>
> Besides animation, other differences lie in canonical avatar creation and scene making. Thanks to our proposed 3D-consistent SDS loss, we could use a more concise pipeline to create canonical avatar without coarse-to-fine and multi-bbox training, achieving faster training speed (only 46.2% training time of AvatarCraft, using one NVIDIA A100 GPU) and comparable visual quality.
>
> We also explore scene making with interactions across avatars and objects (which is not discussed by AvatarCraft), finding that scene-specific fine-tuning with our proposed 3D consistent SDS can further eliminate artifacts and make interactions more realistic.
>
> ### **Q: Why classify AvatarCraft as not animatable in Table 1?**
>
> **A:** We gently remind the reviewer that we classify AvatarCraft as animatable in Table 1, where DreamAvatar is classified as not animatable. Although DreamAvatar allows pose control, the target pose needs to be pre-determined because it requires retraining for each pose control and thus is impractical for animation due to the inefficiency (takes two hours for a novel pose) and inconsistency (the appearance of the avatar in different poses may be changed, as shown in Fig. 1 of the rebuttal pdf). In fact, DreamAvatar doesn't claim to be able to animate avatars, nor does it provide animation results. For the above reasons, we believe DreamAvatar can be safely classified as not animatable.
>
> ### **Q: Using SMPL, DreamWaltz is quite similar to AvatarCraft.**
>
> **A:** Yes, we admit that DreamWaltz can change the pose parameters to implement rough avatar animation, which is quite similar to AvatarCraft. However, DreamWaltz can animate complex avatars like “Woody with cowboy hat”, while AvatarCraft can only animate SMPL-like avatars such as “Woody” with a bald head, as shown in Fig. 7(c) of our paper.
>
> ### **Q: Concern about the term “generalizable NeRF”.**
>
> **A:** We sincerely appreciate the reviewer's suggestion and will change the term "generalizable NeRF" to "deformable NeRF" in the revision.
>
> ### **Q: There exists large improvement room for the current manuscript.**
>
> **A:** We thank the reviewer for the positive comments on the soundness and performance of our work. The current manuscript will be carefully polished. DreamWaltz solves the incompatibility between complex structure (non-SMPL topology) and animation in the existing text-driven avatar creation works, and explores making scenes with diverse interactions. We sincerely hope that our responses could address the reviewer’s concerns.
>
> ### **Q: How to implement avatar-object and avatar-scene interaction?**
>
> **A:** Please refer to Q4 in the global rebuttal for these implementation details.
>
> ### **Q: Missing Limitations and Societal Impacts.**
>
> **A:** We have supplemented the discussion of limitations and societal impacts in the global response.

---

> > ### Comment · Reviewer_kXf3 · 2023-08-18
> >
> > Thanks for the answers. The authors have addressed most of my concerns. However, based on the rebuttal, I still find the avatar-object and avatar-scene interaction not technically challenging enough. So, it is not reasonable to claim these points as the main contribution or novelty. Again, the work itself is good and I believe there exists ample improvement room in the paper's presentation. I will keep my rating based on the current manuscript.

---

> > > ### Author Response · Authors · 2023-08-18
> > >
> > > We appreciate the reviewer's efforts and valuable comments.
> > >
> > >  Making a scene with avatars and interactions is an extremely tricky task due to ambiguous supervision under the zero-shot setting. Our work contributes to **visual quality** (*better than previous text-to-avatar works with shorter generation time*), **animation** (*able to animate complex avatars like "Mobile Suit Gundam" in Fig. 2 of Supp. Material, which has never been shown in previous works*), and explores improvements to **interactions** which are *effective and easy to use (same framework as used for avatar generation)*.
> > >
> > > We thank the reviewer again for the recognition of our work itself, and believe that the deficiencies in the paper's presentation can be resolved in time and will not affect the contributions.

---

### Author Rebuttal · Authors · 2023-08-10

We sincerely thank all reviewers and ACs for their time and efforts. Below we provide the responses to some frequently asked questions and main concerns, as well as the discussions of limitations and societal impacts.

### **Q1: Contribution of our work w.r.t previous methods.**
**A:** Previous methods for text-driven avatar generation include AvatarCLIP [1], AvatarCraft [2], and DreamAvatar [3]. Compared to them, we achieve both complex (non-SMPL-like) and animatable avatar generation for the first time. Besides, we explore how to make scenes with diverse interactions, where our framework can be used for scene-specific fine-tuning to further improve visual quality and interaction realism with 2D prior where social interactions are abundant .

### **Q2: Motivation of our animation learning.**
**A:** Text-driven avatar generation is zero-shot and cannot obtain animation information from video data, so existing works such as AvatarCLIP and AvatarCraft rely on SMPL's inverse skinning for animation. However, SMPL only describes naked human bodies and is not suitable for animating avatars with complex appearances. The core idea of our animation method is to distill pose-dependent appearance knowledge from the pretrained diffusion model for deformation learning, which effectively compensates for the deficiency of SMPL-based animation.

### **Q3: Why is DreamAvatar classified as non-animatable?**

**A:** DreamAvatar allows posed-avatar generation, however this requires retraining (takes about 2 hours) for each new target pose and cannot guarantee a consistent appearance (as shown in Fig. 1 of the rebuttal pdf), making DreamAvatar non-animatable.

### **Q4: How to implement avatar-object and avatar-scene interaction?**

**A:** Implementation details of these two interactions are given as follows. We will add these details in the revision.

**Avatar-object interaction.** The whole training process of avatar-object interaction is actually the same as pure avatar generation, except that the text prompt needs to add object descriptions. For example, we change “Lara Croft” to “Lara Croft with weapons” and change “Kobe Bryant” to “Kobe Bryant with basketball”, obtaining the results in Fig. 1(c) of our paper. In fact, our method can further achieve avatar-object animation, as shown in Fig. 2 of the rebuttal pdf.

**Avatar-scene interaction.** First, we learn the animatable avatar NeRF representation (e.g., “Woody”) via Stage I + II of DreamWaltz, and train the static scene NeRF representation (e.g., “a chair made of cheese”) via Latent-NeRF. Then, both the avatar and the static scene NeRFs are aligned manually and can be rendered by the same camera. We introduce an extra fine-tuning stage as mentioned in sec 3.2.3, utilizing the proposed 3D-consistent SDS loss to fine-tune the hybrid avatar-scene NeRFs and the introduced density weighting network. Here the SDS loss is scene-specific, since we condition the ControlNet on the scene-specific textual description (e.g., “Woody sitting in a chair made of cheese and applauding”) and skeleton images (e.g., random frames from the “sitting and clapping” motion sequence). It takes 30,000 iterations for the extra fine-tuning stage, as described in L78-L79 in the supplementary material pdf.

### **Limitations**
Although DreamWaltz can generate SOTA high-quality complex avatars from textual descriptions, the visual quality can be significantly improved with higher resolution training at higher time and computation cost. The quality of face and hand texture can be further improved through dedicated optimization of close-up views as well as adopting SMPLX instead of SMPL for ControlNet conditioning.

The body portions can be freely controlled via specifying SMPL model parameters, for future work we could train a SMPL shape generative model to facilitate the textual control of SMPL body shapes.

Animation relies on distilling 2D prior depicting randomly selected poses into NeRF representations and the introduced NeRF re-weighting module. Extended training durations tend to yield heightened animation quality and fewer artifacts. However, determining the optimal training duration a-priori remains challenging. To address this issue, we intend to investigate metrics conducive to quantifying training convergence.

Following the optimization-based approach akin to DreamFusion, the generation of an avatar necessitates a dedicated optimization procedure for each textual input, consuming approximately an hour. In the interest of expediting the avatar creation process for arbitrary text inputs, we are poised to investigate techniques such as modulation to foster generalization, thereby enabling swifter avatar acquisition.

### **Societal Impacts**
Societal impact follows prior 3D generative works such as DreamFusion [4] and AvatarCraft.

Given our utilization of Stable Diffusion (SD) as the 2D generative prior, our model could potentially inherit societal biases present within the vast and loosely curated web content harnessed for SD training. We strongly encourage the usage to be transparent, ethical, non-offensive and a conscious avoidance of generating proprietary characters.

It is important to acknowledge that generative models such as ours may have implications for the displacement of creative workers through automation.  Nevertheless, DreamWaltz is intended to be a tool to liberate designers and animators from laborious and repetitive work, thereby to focus on intellectual creativity and to enhance accessibility.

### **Reference**

[1] AvatarCLIP: Zero-Shot Text-Driven Generation and Animation of 3D Avatars. SIGGRAPH 2022.

[2] AvatarCraft: Transforming Text into Neural Human Avatars with Parameterized Shape and Pose Control. ICCV 2023.

[3] DreamAvatar: Text-and-Shape Guided 3D Human Avatar Generation via Diffusion Models. Arxiv 2023.

[4] DreamFusion: Text-to-3D using 2D Diffusion. ICLR 2023.

---

### Decision · Program_Chairs · 2023-09-21

**Decision:**

Accept (poster)

**Comment:**

This paper proposes a method for creating 3D human avatars from text prompts using guidance from a pre-trained diffusion model. The main claimed contributions of this work are (a) improvements in visual quality (b) ability to create posed characters and (c) the ability to create characters with objects and in scenes.

The reviewers' scores hover around borderline (3BR, 1BA).

The first main legitimate concern that multiple reviewers raised was the fact that the claimed contribution around being able to generate scenes with multiple interacting characters or objects and characters interacting with each other, the technical details of this part are not well-described in the paper. The authors described it in much detail in their response to the reviewers. However, the approach that they used to create these scenes is simplistic -- render two characters/an object and a character separately first with their individual text prompts, rigidly align them and then fine tune their Nerfs together with a text prompt. For characters holding objects they used text prompts of the type "[X] holding [Y]". So, in essence, there is nothing unique about the proposed method that enables this capability. Hence, the AC does not regard this a compelling contribution of this work and strongly recommends that the authors to significantly tone down this claimed contribution in the final manuscript and simply show it as an additional application/capability in the results section.

The second major concern of the reviewers' is around comparisons to the works AvatarCraft [2], and DreamAvatar [3], both in terms of conceptual novelty and the quality of the results. As per the NeurIPS guidelines, the authors are not required to compare to the concurrent works of AvatarCraft [2], and DreamAvatar [3]. Hence, the AC and SAC disregarded them while making the final decision.

Without AvatarCraft [2], and DreamAvatar [3], in the picture, the proposed work shows a clears advancement over the state-of-the-art prior methods in terms of quality, functionality (models articulation) without requiring optimization for each pose.

Hence, the final decision is to accept this work. Congratulations!

The authors are strongly advised to make changes to the original manuscript that they've promised to the reviewers, add clear details of how "scenes" with multiple avatar and/or objects are created and tone down this claimed contribution.